# Molecular basis of ion permeability in a voltage-gated sodium channel

Claire E Naylor[1,†], Claire Bagnéris[1,†], Paul G DeCaen[2,3], Altin Sula[1], Antonella Scaglione[2,3,‡], David E Clapham[2,3] & BA Wallace[1,*]

## Abstract

**Voltage-gated sodium channels are essential for electrical signalling across cell membranes. They exhibit strong selectivities for sodium ions over other cations, enabling the finely tuned cascade of events associated with action potentials. This paper describes the ion permeability characteristics and the crystal structure of a prokaryotic sodium channel, showing for the first time the detailed locations of sodium ions in the selectivity filter of a sodium channel. Electrostatic calculations based on the structure are consistent with the relative cation permeability ratios ($Na^+ \approx Li^+ \gg K^+$, $Ca^{2+}$, $Mg^{2+}$) measured for these channels. In an E178D selectivity filter mutant constructed to have altered ion selectivities, the sodium ion binding site nearest the extracellular side is missing. Unlike potassium ions in potassium channels, the sodium ions in these channels appear to be hydrated and are associated with side chains of the selectivity filter residues, rather than polypeptide backbones.**

**Keywords** crystal structure; electrophysiology; ion permeability; sodium channel

**Subject Categories** Membrane & Intracellular Transport; Structural Biology

**The EMBO Journal (2016) 35: 820–830**

See also: **W Kühlbrandt** (April 2016)

## Introduction

Voltage-gated sodium channels are essential for electrical signalling across cell membranes, and their opening initiates the action potential in excitable cells. Mutations in human sodium channels result in a wide range of neurological and cardiovascular channelopathies, and as a consequence, they are important targets for pharmaceutical drug design. Sodium channels enable the translocation of sodium ions across cell membranes with exquisite selectivity over other types of cations, thus permitting the specific cascade of events associated with electrical signalling in cells. Eukaryotic sodium channels ($Na_v$s) consist of pore-forming α subunits, which alone are sufficient for functional expression of the channel properties (Noda et al, 1986). They are single polypeptide chains of ~2,000 residues that contain four homologous repeats, each of which comprises a structural domain (designated DI-DIV), where each domain consists of six transmembrane helices, designated S1–S6. The four domains assemble to form a pseudotetramer, with the S5 helices, intervening P-regions, and S6 helices comprising the pore subdomain that forms the pathway for the translocation of ions across the cell membrane. The S1–S4 helices form the voltage sensor subdomains. No crystal structures have yet been determined of any eukaryotic sodium channel. However, bacteria possess simpler versions of voltage-gated sodium channels (Ren et al, 2001) that have many functional properties that are similar to those of eukaryotic sodium channels, including selectivity for $Na^+$ ions (Ren et al, 2001; Irie et al, 2010; DeCaen et al, 2014), and, for the NavMs orthologue studied here, binding of local anaesthetics and channel-blocking drugs (Bagnéris et al, 2014). Prokaryotic sodium channels are homotetramers, in which each monomer contains six transmembrane segments and is homologous to a single eukaryotic pseudorepeat. As with the eukaryotic channels, their S1–S4 segments form the voltage sensors, and the S5–S6 segments from each polypeptide chain associate to form an eight-helix bundle that constitutes the pore. The selectivity filter (SF), which determines the channel's $Na^+$ permeability over other cations, is formed by the close association of the P loops from each of the polypeptide chains.

The structures of several prokaryotic sodium channels in different functional states (Payandeh et al, 2011, 2012; McCusker et al, 2012; Zhang et al, 2012; Bagnéris et al, 2013, 2014, 2015; Tsai et al, 2013; Shaya et al, 2014) have been solved by X-ray crystallography or electron microscopy. Functional studies on wild-type and mutant prokaryotic channels (Ren et al, 2001; Yue et al, 2002; McCusker et al, 2011; Shaya et al, 2011; Tang et al, 2014) have suggested which regions and residues may be important for ion binding, selectivity and translocation within the SF. However, the detailed locations and environments surrounding the sodium ions within the channels have not been discernable

1  Institute of Structural and Molecular Biology, Birkbeck College, University of London, London, UK
2  Department of Cardiology, Howard Hughes Medical Institute, Boston Children's Hospital, Boston, MA, USA
3  Department of Neurobiology, Harvard Medical School, Boston, MA, USA
   *Corresponding author. Tel: +44 207 6316800; E-mail: b.wallace@mail.cryst.bbk.ac.uk
   †These authors contributed equally to this work
   ‡Present address: Department of Biochemical Sciences, Institute of Molecular Biology and Pathology of CNR, Sapienza University of Rome, Rome, Italy

   

from the crystal structures published to date. Whilst the locations of sodium ions were proposed based on the structure of the closed pore form (Payandeh *et al*, 2011), no density was seen in the SF in those crystals to indicate any sodium ions were present. Only structures of the open state (McCusker *et al*, 2012; Bagnéris *et al*, 2014) pore of NavMs from *Magnetococcus marinus* have shown any electron density within the SF that might be attributable to sodium ions. The resolution of the first such structure (3.5 Å), however, was too low to determine the details of the sites of the ions in the SF, whilst the latter structures contained channel blocker ligands which interfered with ion binding. The sodium ion binding site locations and mechanisms for ion selectivity have also been predicted by a number of molecular dynamics (MD) studies, although they differ considerably in their details (Amaral *et al*, 2012; Chakrabarti *et al*, 2013; Stock *et al*, 2013; Ulmschneider *et al*, 2013; Xia *et al*, 2013; Zhang *et al*, 2013; Boiteux *et al*, 2014; Furini *et al*, 2014).

In this study, we analysed new crystals of the NavMs pore that contain only sodium ions and diffract to higher resolution (2.7 Å), enabling us to identify the sodium binding sites in the SF. We have also examined the relative permeabilities of different monovalent and divalent cations in NavMs channels expressed in HEK293T cells; these results correlate well with electrostatic calculations of different ion profiles, based on the crystal structure. In addition, we have mutated the glutamic acid (residue 178) found at the top of the SF, in the position corresponding to the DEKA SF motif present in eukaryotic sodium channels; this removed the first (most extracellular) ion binding site and reduced the permeability and selectivity of the channel for sodium over calcium and other cations.

# Results

## Electrophysiological measurements of cation selectivities

The NavMs channel has previously been shown to exhibit a significant selectivity for $Na^+$ over $K^+$ (Ulmschneider *et al*, 2013). In this study, whole-cell patch clamping was used to determine the relative permeabilities of additional monovalent and divalent cations ($Li^+$, $Cs^+$, $Ca^{2+}$, $Mg^{2+}$ and $Ba^{2+}$) in the NavMs channel (Fig 1 and Appendix Fig S1A, Table 1A). Voltage-dependent inward currents were only observed in the presence of extracellular sodium or lithium ions (Fig 1A). To exclude the possibility of anomalous mole fraction effects created by the internal sodium, sodium was replaced by the impermeant cation, N-methyl-D-glucamine (NMDG) from the intracellular saline in other experiments (Appendix Fig S1). As observed in the case with internal sodium, only $Li^+$ and $Na^+$ currents could be measured (Appendix Fig S1B and C, Table 1B). No voltage-gated current could be measured in the presence of $Cs^+$, $Ca^{2+}$, $Mg^{2+}$ or $Ba^{2+}$ (Appendix Fig S1C, Table 1B) in either the presence or absence of internal sodium ions. Comparisons of the expected-to-measured reversal potentials for Na:Li give an estimated relative permeability ratio of 1:0.8. The high values for $Na^+$ and $Li^+$ over other monovalent and divalent cations determined in this study for NavMs are similar to values that have been measured for eukaryotic sodium channels, such as rat muscle $Na_v1.5$ [$P_{X/Na} = 1$ for $Li^+$ and $< 0.08$ for $K^+$, $Cs^+$, $Ca^{2+}$ and $Ba^{2+}$ (Sun *et al*, 1997)].

## Ion binding sites

The NavMs sodium channel pore protein was purified and crystals grown in the presence of sodium ions (Appendix Table S1). The pores in these crystals are in the open conformation, with the activation gate at the intracellular ends of the transmembrane domains of all four monomers splayed to allow ions to exit the pore, as in Bagnéris *et al* (2013). The crystals include two crystallographically distinct tetramers ("AB" and "CD"), for which root mean square deviations (RMSDs) between all atoms and SF-only atoms in the optimally superposed tetramers are only 0.802 and 0.278 Å, respectively, suggesting that the structures of the polypeptides in both tetramers are very similar. Therefore, for clarity, we will focus our discussion on the structure of the AB tetramer.

The locations of three highly occupied sodium ion sites are seen in the centres of the selectivity filters (Fig 2A), with ion B-factors similar to those of the protein atoms (Appendix Table S1). The crystallographic identification of these sites as arising from sodium ions is discussed in detail in the Appendix (including Appendix Figs S2–S6). Densities attributable to partially occupied water molecules are distinct (Appendix Fig S2F) and can also be seen within the SF (Fig 2) and at the bottom of the SF/top of the hydrophobic internal cavity close to the local anaesthetic channel blocker binding sites (Bagnéris *et al*, 2014). The sodium site nearest to the extracellular surface ($Na_I$; top cyan ball in Fig 2A) is located close to the site that had been designated S1 (Fig 2B) in molecular dynamics studies of the NavMs pore (Ulmschneider *et al*, 2013). This corresponds to the $Site_{HFS}$ (Fig 2B), proposed based on the apo-ion closed structure of the NavAb channel (Payandeh *et al*, 2011), which has the same SF sequence as NavMs (LESWSM). This site is nearest to the carboxylate group of SF residue Glu178 and the hydroxyl group of SF residue Ser179. In the MD simulations, the S1 sodium ion is asymmetrically disposed with respect to the SF, but due to the symmetry of the tetramer in the crystal, this is consistent with the observed slightly wider (but not asymmetric) appearance of the highest level contour. Below that is the site we designate as $Na_{II}$, which is closest to the carbonyl oxygen of Leu177, and is separated from the $Na_I$ site by ~2.8 Å. It is located close to, but not coincident with, the low-occupancy S2 site proposed from MD calculations (Ulmschneider *et al*, 2013) and above the $Site_{CEN}$ proposed from the apo-structure (Payandeh *et al*, 2011). $Na_{III}$ corresponds closely to the site denoted S4 in the MD studies and above the proposed $Site_{IN}$. It is nearest to the carbonyl oxygen of Thr176; it is separated from $Na_{II}$ by ~5.6 Å. There is no site equivalent to the S3 site identified by MD calculations (which was unlikely to be occupied at the same time as S4 for steric reasons), nor was there any density visible in the low-occupancy S0 site external to the SF that was proposed based on the MD simulations. These differences between experimental and hypothesised or simulation results are significant and are why the availability of actual experimental results is valuable for understanding the molecular basis of the ion permeability in these channels. There is an additional peak on the symmetry axis in the AB tetramer between sites S2 and S3, which has been assigned as being water but cannot be absolutely discounted as being a lower occupancy sodium ion. The more conservative interpretation has been to model this density as water, as the electron density at this location is at lower than that for the sodium ion sites and only appears at low sigma values in omit maps (Appendix Fig S5). However, it is

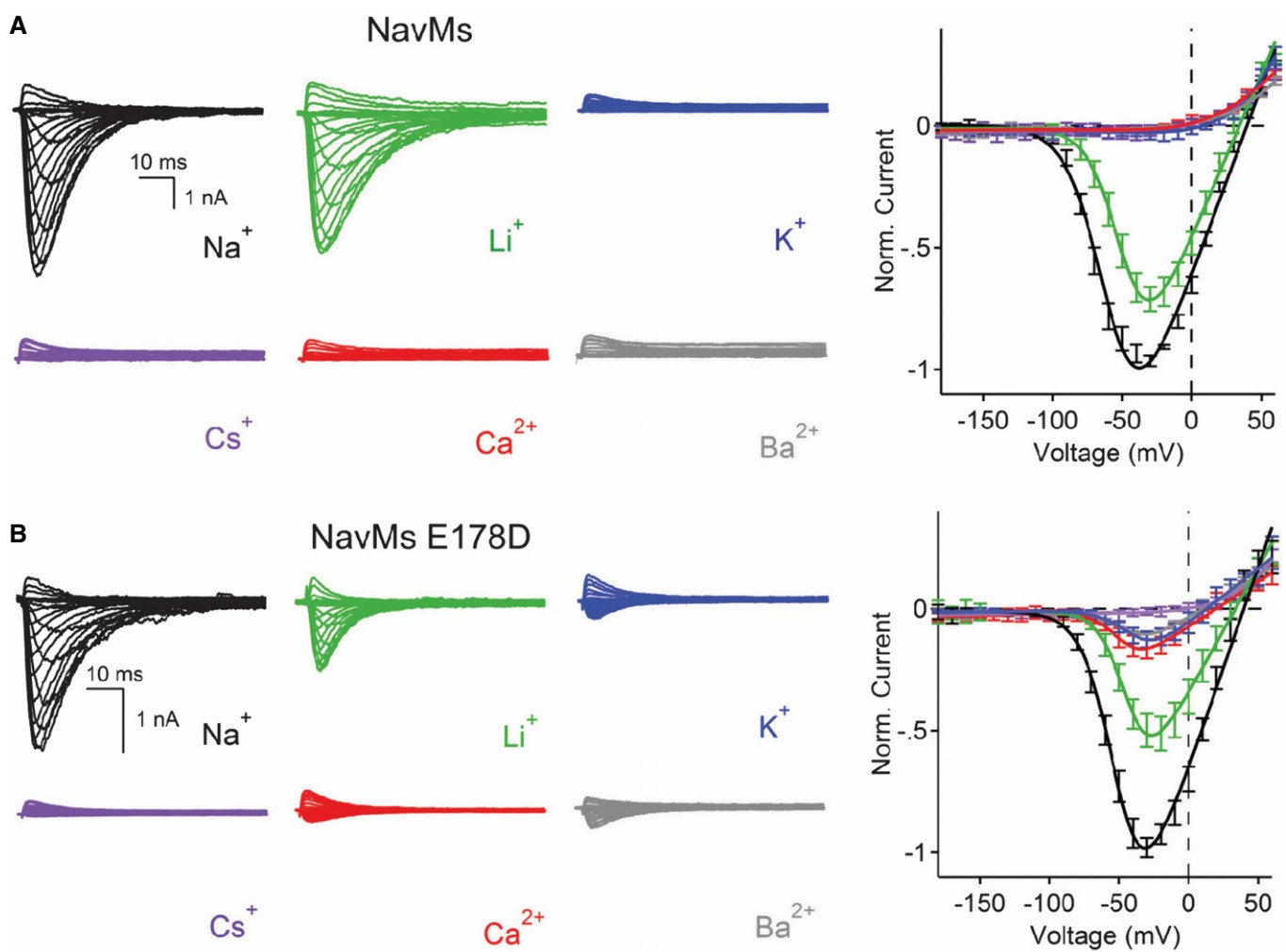

**Figure 1. Patch-clamp measurements of the ion selectivity of wild-type and E178 mutant NavMs channels.**

A Left: Representative current traces from wild-type NavMs channels showing the first 0.1 s of 0.5 s activations from −180 mV holding potential recorded with the indicated extracellular cations. Right: Resulting current–voltage profiles, where current magnitudes were normalised to the maximum inward current measured in the sodium condition (error ± SEM; *n* = 6–9).

B Left: Representative current traces from mutant channel E178D (as in A). Right: Resulting current–voltage profiles (error ± SEM; *n* = 4–8).

**Table 1. Measured reversal potentials ($E_{rev}$) and relative ion permeabilities ($P_x$:$P_{Na}$) of the NavMs channel for different cations. (A) With 30 mM internal sodium present (see Materials and Methods for detailed description of saline conditions). (B) Monovalent cations were tested with matching 5 mM internal monovalent ions. Divalent cations were tested with internal saline that contained ≈ 1 μM free-Ca$^{2+}$ (see Materials and Methods for detailed description of saline composition).**

| Channel/Ion | $E_{rev}$(in mV)\ $P_x$:$P_{Na}$ | | | | | |
|---|---|---|---|---|---|---|
| | Li$^+$ | Na$^+$ | K$^+$ | Cs$^+$ | Ca$^{++}$ | Ba$^{++}$ |
| A | | | | | | |
| Wild-type | 32 ± 2\0.7 | 36 ± 2\1.0 | −1 ± 3\< 0.2 | −1 ± 2\< 0.2 | −4 ± 2\< 0.2 | −7 ± 3\< 0.2 |
| E178D | 33 ± 2\0.7 | 37 ± 2\1.0 | 14 ± 2\0.4 | −1 ± 3\< 0.2 | 16 ± 2\0.5 | 11 ± 3\0.3 |
| B | | | | | | |
| Wild-type | 67 ± 3\0.8 | 80 ± 3\1.0 | −3 ± 3\< 0.04 | −2 ± 3\< 0.04 | < −3 ± 2\< 0.04 | < −2 ± 3\< 0.04 |

possible this site is occasionally occupied by a sodium ion, and in that case, there could actually be four sodium ions rather than three in the SF.

An important finding is that there are no protein atoms within the normal outer limit (3.5 Å) for a sodium ion coordination sphere (the closest is Oγ1 of Ser179 at 3.79 Å from Na$_I$), suggesting that

protein–ion interactions are mediated by intermediate water molecules. Ordered or partially ordered water molecules are observed in the lower part of the SF, near sites $Na_{II}$ and $Na_{III}$ (compare Fig 2 and Appendix Fig S2F), at appropriate Na-O distances.

Whilst the $Na^+$ sites in the CD tetramer are similar to those in the AB tetramer, there are small differences (on the order of 0.2–0.4 Å) in their positions (Fig 2B), despite the structures of the polypeptide backbones being so similar. These differences are not concerted, however, and result in $Na_{II}$ and $Na_{III}$ being ~0.5 Å closer in the CD tetramer (placing $Na_{III}$ nearer to the S3 MD position), whilst $Na_I$ and $Na_{II}$ are 0.2 Å further apart in that tetramer. Thus, these slightly different sodium positions in the two nearly identical AB and CD tetramers suggest that sodium ions are capable of binding to several closely related sites as they pass through the SF, possibly due to the presence of intervening water molecules along the length of the SF (including ones that are not sufficiently well ordered to be seen in the crystal structure) and the lack of direct interactions with the SF polypeptides.

## Electrostatics calculations of cations binding in the SF

Electrostatic potential calculations (Appendix Fig S7), using this NavMs pore crystal structure and monovalent cations with a range of radii, indicate that monovalent ions with the same radius as sodium, and those with smaller radii (e.g. lithium) have favourable interactions that would enable them to pass through the SF, as observed in the conductance studies. The experimental binding sites seen in the SF are located in a flat and somewhat shallow but favourable binding region for cations. The flatness of this region may reflect the modest variability seen in the sodium binding sites in the AB and CD tetramers. The plots suggest that larger monovalent ions would be excluded at the narrowest point of the selectivity filter, close to where the Glu178 side chain extends into the channel (Appendix Fig S7A), again consistent with the electrophysiology measurements. As sodium is the smallest monovalent ion commonly found in biological systems, the exclusion of only larger ions is an effective strategy for producing monovalent ion selectivity. Dehydrated potassium ions are approximately the right size to pass through the channel, but the small interaction energies with the polypeptide would not enable compensatory desolvation (as occurs in potassium channels), and $K^+$ thus does not pass through the NavMs SF.

Similar calculations with divalent cations (Appendix Fig S7A) show repulsion due to their higher charge, which means that even divalent ions with small radii (such as calcium or magnesium, which have a similar radii to sodium) would be excluded at the entry of the SF. There also appears to be an additional repulsion of divalent ions lower in the pore due to the hydrophobic side chain of Ile215 that protrudes into the channel near where the pore narrows at the bottom of S6.

Hence, these calculations based on the crystal structure correlate well with the functional measurements of ion selectivity.

## Influence of SF sequence on ion permeability, selectivity and structure

The ion selectivity profiles for other prokaryotic sodium channels have been altered by the mutation of the SF residues (Yue *et al*,

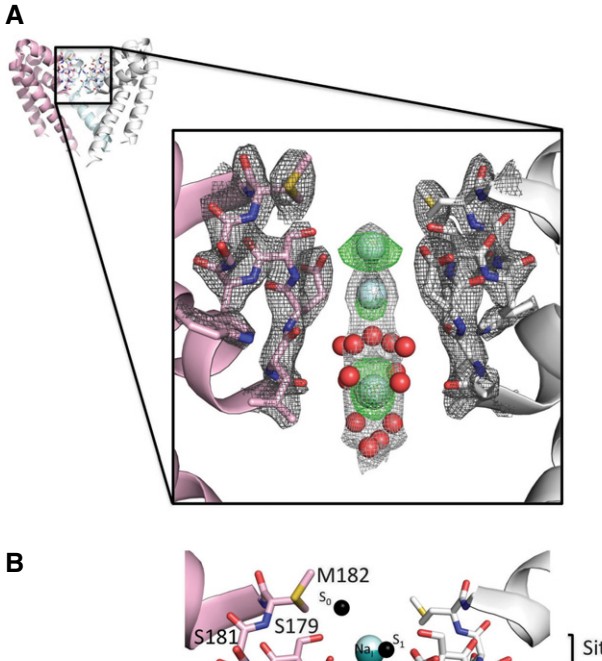

**Figure 2.  Crystal structure (PDBID 5BZB) showing sodium ion binding sites and ordered waters in the wild-type NavMs pore.**

A  Upper: Overview of the AB tetramer, with each of the monomers depicted in a different colour. The front monomer has been removed for clarity. The black box indicates the region expanded in the lower panel. Lower: An expanded view of the selectivity filter, showing only two monomers as pink and white ribbon cartoons with the selectivity filter residues drawn as sticks in the corresponding colours. Sodium ions are shown as cyan spheres and well-ordered waters as red spheres (at 0.4× their actual radii, for clarity). The final refined $2F_o$-$F_c$ electron density maps for the protein, ions and water are shown at 2.0 σ in dark grey mesh and 0.8 σ in light grey, and the initial difference map (without sodium or waters included in the phase calculation) at 3.0 σ in green mesh.

B  Superposition of the sodium ion binding sites found in the crystallographically distinct AB and CD tetramers and the proposed binding sites from other studies. The tetramers were superposed with the LSQKAB program (Kabsch, 1976) from CCP4 (Winn *et al*, 2011) using residues 131–220 from all four monomers in each tetramer. Only two monomers from the AB tetramer are shown for clarity, and coloured as in (A). Sodium ions from the AB tetramer are depicted as cyan spheres, those from the CD tetramer as teal spheres. Predicted sodium ion binding sites from Ulmschneider *et al* (2013) are indicated as small black spheres and labelled on the right of the spheres according to the nomenclature ($S_0$–$S_4$) in that study. Predicted sodium ion binding sites from Payandeh *et al* (2011), based on a closed channel structure of the close orthologue NavAb, are indicated on the far right and labelled as per the nomenclature of that study ($Site_{HFS}$, $Site_{CEN}$ and $Site_{IN}$). This and subsequent molecular figures were drawn with PyMol (The PyMOL Molecular Graphics System, Version 1.6, Schrödinger, LLC) unless otherwise stated.

2002; Shaya *et al*, 2011; DeCaen *et al*, 2014; Finol-Urdaneta *et al*, 2014; Tang *et al*, 2014). In the NaChBac orthologue, mutating the SF sequence from LESWAS to LEDWAS made these channels

non-selective, and mutation to LDDWAD enhanced calcium selectivity (Yue et al, 2002). Mutation of the NavSp SF from LESWSM to LDDWSD increased its affinity for calcium, although the channel still preferentially conducted sodium over potassium (Shaya et al, 2011); the same changes to the NavAb SF (from LESWSM to LDDWSD) produced a calcium-selective channel (Tang et al, 2014). The effects of these changes correlate well with measured selectivities of other naturally occurring prokaryotic channels that have different SF sequences (DeCaen et al, 2014).

To specifically test the role of the E178 residue located at the extracellular end of the SF in the selectivity and translocation properties of NavMs, we mutated its SF from LESWSM to LDSWSM (Fig 1B, Appendix Fig S8). This mutation changes the residue near the $Na_I$ site from a glutamic acid to the shorter aspartic acid, but does not alter the net charge of the SF. The current density of the E178D mutant channel was 15-times less than that found in the wild-type channel measured under sodium-only conditions (11.4 ± 3 vs. 165 ± 21 pA/pF). As the structure demonstrates that there is one less binding site in the mutant channel than in the wild-type structure, this could be consistent with fewer ions passing through the channel per unit of time, and therefore a lower current density. Electrostatic calculations (Appendix Fig S7B) suggested this should produce a less-selective channel, with reduced barriers to $K^+$ and $Ca^{2+}$ at the entry to the SF (but unchanged at the lower site), consistent with the lower relative selectivity ratios for $Na^+$ over $K^+$ and $Ca^{2+}$ (Table 1B) measured for the mutant. $K^+$, $Ca^{2+}$ and $Ba^{2+}$ conductances were detected (albeit at lower levels) in addition to $Na^+$ and $Li^+$ (Fig 1B). Thus, the glutamate at this position within the NavMs filter appears to be essential to achieve the high level of sodium selectivity seen in native channels.

The single-channel $Na^+$ conductance of E178D was half that of the wild-type NavMs channel (γ = 17 ± 2 vs. 33 ± 3 pS), and was blocked by internal application of the charged lidocaine derivative QX-314 (Appendix Fig S8). However, the relative open probabilities of the wild-type and mutant cells were not significantly different. Thus, at least half the observed reduction in current density is due to a reduction in $Na^+$ conductance. The remaining difference between wild-type and E178D current density may be due to alterations to membrane trafficking of the mutant channel producing differences in the number of functional channels assembled on the plasma membrane. Additional mutations at positions S181 and M182 within the selectivity filter (LESWGM; LESWGS; LDSWGS; LDSWDM; and LESWSD, where mutation sites are underlined) were made but none produced channels with functional currents, suggesting that NavMs is less tolerant than the NaChBac and NsvBa orthologues to amino acid substitutions at extracellular filter sites S181 and M182 (DeCaen et al, 2014).

The crystal structure of the E178D mutant (Appendix Table S1, Fig 3) showed considerable differences from the wild-type channel local to the site of the mutation: the methylene group of the glutamic acid was missing, producing a side chain that projected to a lesser extent into the SF, as expected (Fig 3A), and the site at $Na_I$ was completely devoid of ordered electron density, suggesting that it was unoccupied by an ion (Fig 3A right). Despite these differences in the side chain densities at the site of the mutation and adjacent sodium ion, the all-atom protein RMSD between the wild-type and mutant structures is only 0.778 Å (and just 0.478 Å over the 40 residues (150–190) in the external leaflet of the pore (tops of S5 and S6

and the intervening SF and pore helices which make up the selectivity filter and its surroundings protein residues), indicating the mutation produces a very local change, making it a good validation test for the effects of only that one residue on the ion binding.

Soaking the crystals in calcium, which might have been expected to produce additional density if the SF had become calcium-selective, produced a similar result: no density was seen in the region of the D178 residue (data not shown). Relative to wild-type (Fig 3B left), the mutant has a more electropositive region (denoted by the circle in Fig 3B right) at the top of the SF, due to the exposure of the backbone nitrogen of residue S179 when the longer side chain of residue E178 is replaced with the shorter side chain in the mutant; this could tend to inhibit cations from binding at this site, consistent with measured decreased current densities. It is notable that $Na_v$s, which are pseudotetramers formed from single polypeptide chains, have asymmetric SFs, where the equivalent residues to E178 in the four domain motif are DEKA (except in aphids, where apparently the motif is DENS (Amey et al, 2015)), thereby providing both a short and long acidic side chain at this position in the different domains.

### Comparison of sodium ion positions in the NavMs pore with calcium ion positions in the NavAb channel redesigned to be a "Calcium Channel"

The NavAb channel has been engineered to make it more $Ca^{2+}$-selective (Tang et al, 2014) (designated CavAb) by substitution of three negatively charged amino acids in the SF (LESWSM mutated to LDDWSD). Unlike the wild-type open-pore NavMs structure, the wild-type closed-pore NavAb structure did not contain any visible ion density in the SF. The crystals of the CavAb were grown in the same way as wild-type NavAb, but later soaked in solutions containing calcium. Although the mutant structure was still closed at the pore egress, calcium ions (identified by their anomalous scattering signals) were observed in the selectivity filter of this engineered channel.

As observed in the present study for the wild-type NavMs pore containing sodium ions, no direct interactions between calcium ions and SF residues were seen in the NavAb structure. In the CavAb SF, the ions appeared to be hydrated. The protein atoms closest to the hydrated calcium ions are identical to those seen interacting with sodium in the wild-type NavMs sodium pores in this study (Appendix Fig S9A): the carboxyl oxygen of the side chain of Asp178 (numbering according to NavMs sequence). The introduced additional negative charges (in the side chains mutated to Asp) did not interact directly with the ions in CavAb, but rather (based on distance measurements) appeared to be mediated by intervening water molecules. The multiple mutations introduced in the CavAb channels generate large electrostatic interactions with cations (Appendix Fig S9B), a force which is correspondingly larger for the more highly charged calcium ions than for sodium or potassium ions. This thus appears to allow ions to bind in the sodium ion binding sites despite the channel being closed, overcoming even the unfavourable interactions at the intracellular end of the pore. It is notable that the locations of the CavAb calcium ions are similar to the locations of sodium ions in the AB tetramer of the wild-type NavMs selectivity filter (Appendix Fig S9A). These differ somewhat from the previously predicted sites for sodium ions.

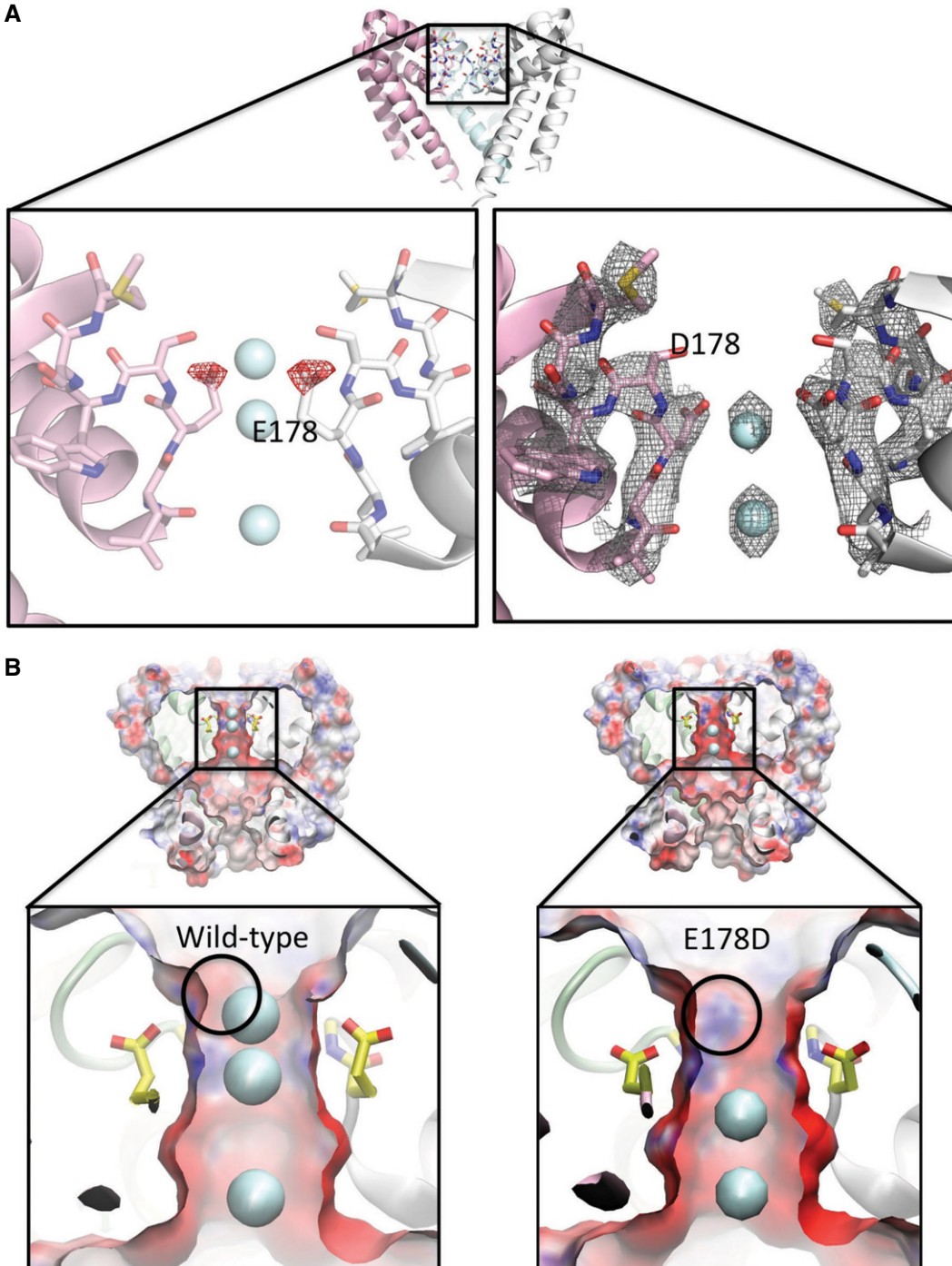

**Figure 3.  The structure of the E178D selectivity filter mutant.**

A   Top: Crystal structure overview of the wild-type AB tetramer (PDBID 5BZB), view as in Fig 2A. Left: Difference ($F_o–F_c$) electron density map (mutant minus wild-type) contoured at −2.5 σ and drawn as red mesh, plotted on the structure of the wild-type (E178) protein, highlighting the consequences of the loss of the δ-carbon atom from residue 178 in the E178D mutant. For reference, the locations of the sodium ion sites present in the wild-type (E178) structure are depicted as cyan spheres. Both the structure and the ions are displayed as faded backgrounds so that the difference map can be seen more clearly. Right: Final refined $2F_o–F_c$ electron density map for the E178D mutant (PDBID 4X88), contoured at 1.5 σ and shown as grey mesh, with the sodium ions depicted as cyan spheres.

B   Surface representations of (left) wild-type NavMs pore and (right) the E178D mutant coloured by electrostatic charge calculated by APBSmem (Callenberg *et al*, 2010) and drawn with VMD (Humphrey *et al*, 1996), showing a slice through the middle of the tetramer. Sodium ions are shown as cyan spheres; only 2 are present in the mutant, instead of the 3 seen in the wild-type, with the top sodium missing in the mutant. The lower panels are expanded views of the selectivity filter regions, with the E178 side chains depicted in yellow sticks and the location of the main chain nitrogen of residue 179 indicated by the black circle. The alteration in the protein structure caused by removal of the methylene group of E178 exposes a positive region of the polypeptide backbone adjacent to the site where the top sodium ion is located in the wild-type protein; this likely contributes to why an equivalent sodium ion is not seen adjacent to this position in the mutant.

## NavMs

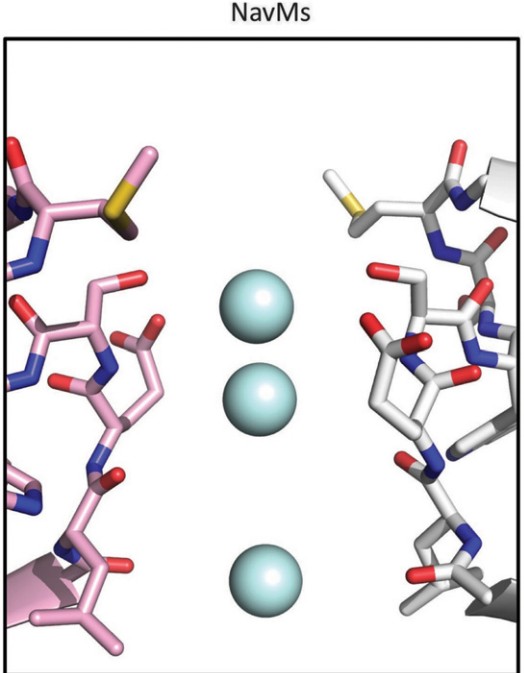

## KcsA

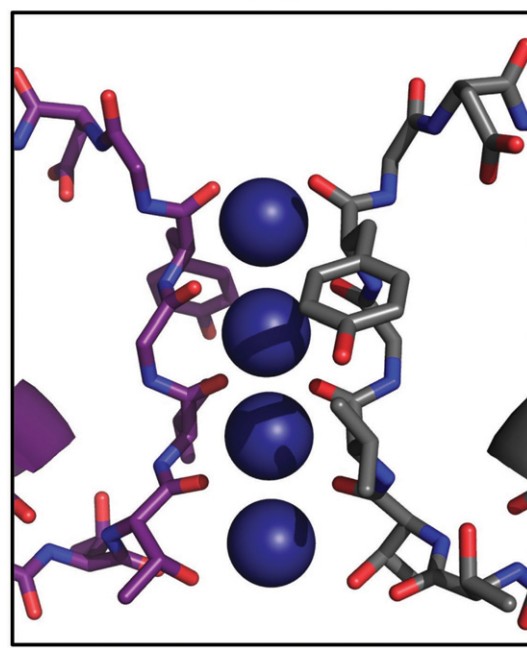

**Figure 4.  Comparison of ion binding in sodium and potassium channels.**
Left: Sodium ions (cyan) in the selectivity filter of the NavMs sodium channel. Only two monomers (in white and pink) are shown for clarity. Coordinates are from the AB tetramer of PDBID 5BZB. Right: Potassium ions (blue) in the selectivity filter of the KcsA potassium channel. As for the NavMs structure, only two monomers (in grey and maroon) are shown for clarity. Coordinates are from PDBID 1K4C (Zhou *et al*, 2001). The two crystal structures were first superimposed using SSM (Krissinel & Henrick, 2004), and are drawn using the same scale, and with the correct ratio between the diameters of the ions used, so that the differences in the internal dimensions of their selectivity filters can be seen, thus making it clear how much more space there is available in the NavMs SF for intervening water molecules. In the KcsA channel, the ions are directly coordinated to the polypeptide backbone atoms, whereas in the NavMs SF, the closest atoms (side chains) are obviously at too great a distance to allow direct bonding to the ions, enabling interactions via water molecules.

## Discussion

The binding sites for sodium ions in the prokaryotic sodium channels described in this study are very different from those found for $K^+$ in either prokaryotic or eukaryotic potassium channels—in number, location and type of coordination. In potassium channels, the binding sites are formed by polypeptide backbone carbonyl groups (Doyle *et al*, 1998), whereas in the NavMs sodium channel the SF is lined with amino acid side chains (Fig 4). Potassium ions are fully dehydrated in potassium channel SFs and form direct contacts with the polypeptide. Sodium ions, on the other hand, retain full or partial hydration in the sodium channel SF and do not come in direct contact with the polypeptide backbone or side chains.

Whilst, in general, different members of the voltage-gated ion channel family have very similar overall structures/folds, their architectures around their SFs are very different. Potassium channels have a very narrow tunnel forming direct connections to the $K^+$ through four backbone carbonyls, with no intervening water molecules. Apparently only two sites out of the four that have been identified can be occupied at any one time (Zhou *et al*, 2001; Jiang *et al*, 2002, 2003). In contrast, the prokaryotic sodium channel SF is wider and its mouth contains a highly conserved glutamate that would allow a hemi-hydrated sodium ion to pass through. Potassium ions, which would also need to be at least partially desolvated in order to fit into the NavMs SF, are

prevented from passage, presumably due to the energy cost of desolvation. Because the contacts between the sodium channel polypeptides and ions must occur via intervening water molecules, the binding sites are less localised than in potassium channels, and would enable rapid passage. The electrostatic nature of the SF is such that it favours the selectivity of the channel for $Na^+$ over $Ca^{2+}$, even though the divalent ion could physically fit within it. Whether or not the selectivity filters of eukaryotic and prokaryotic Navs use similar physical features to achieve sodium selectivity is an important question to be determined in future structural studies of eukaryotic Nav channels.

## Materials and Methods

### Protein expression, purification and crystallisation

pET15-NavMs-Pore-FL expressing the wild-type pore of NavMs from *M. marinus* MC-1 with its complete C-terminal domain has previously been described (Bagnéris *et al*, 2013). The E178D mutant was generated using the QuickChange™ mutagenesis kit (Agilent Technologies). Wild-type and E178D proteins were purified and crystallised according to Bagnéris *et al* (2013, 2014), except the protein preparations were not pretreated with thallium nitrate (i.e. they only contained sodium ions). The differences that

enabled the improvement in resolution were small but manifold, as is often the case for membrane proteins and included such things as very minor changes to the solubilisation and thrombin cleavage procedures and small changes to the protein and crystal handling procedures. As previously noted (McCusker *et al*, 2012), we were unable to grow any crystals under any similar conditions in the absence of sodium ions or when the sodium salts were replaced with other monovalent salts (such as lithium) in the crystallisation condition. Large crystals (above 100 μm) were soaked with 5 and 50 mM ion solutions (made using 100 mM and 1 M stock solutions containing 100% DMSO for cadmium (II) chloride, thallium (I) nitrate, manganese (II) chloride, silver (I) nitrate, or 10 mM Tris, 100 mM NaCl, 0.52% Hega10 [gel filtration buffer] for barium (II) acetate, lithium (I) chloride and calcium (II) chloride), with the aim of replacing the sodium ions with other monovalent or divalent ions to enable the detection of anomalous difference signals. In no instances were the sites internal to the SF replaced by the other ions.

## Crystal data collection, processing, refinement and display

Multiple data sets were collected for each crystal type at beamlines IO4 or IO4-1 (Diamond Light Source, UK), beamline PROXIMA1 (Soleil, France) and beamline ID23-1 (ESRF, France), and the single highest resolution diffracting crystals were used for determining the structures. All data were indexed and integrated with XDS (Kabsch, 2010) and scaled with Aimless (Evans, 2006). All subsequent data analyses were carried out with the CCP4 suite (Winn *et al*, 2011). Diffraction extends to ~2.4 Å resolution, but the crystals suffer from a layer lattice disorder (as discussed in Bagnéris *et al*, 2013) as a result of the disordered C-terminal domain, which is reflected in decreased CC1/2 values as well as raised $R_{merge}$ and $R_{pim}$ values for a given I/σI value (Appendix Table S1). Because of this constraint, a conservative resolution cut of 2.7 Å was used for refinement. The structures were solved by molecular replacement and refinement in Buster (Bricogne *et al*, 2011) using PDBID 3ZJZ (Bagnéris *et al*, 2013) with sodium ions and water removed to prevent bias in the electron density in the filter. Subsequent all-atom refinement was restrained to a 2.2 Å resolution refined structure of the pore in complex with a channel blocker (C.E. Naylor, C. Bagnéris unpublished data). The four identical peptide chains (arranged in two half-tetramers with the tetramers completed by crystal symmetry) were restrained by non-crystallographic symmetry between all four chains initially and at the end of refinement, following tests for optimum refinement protocol, only between chains in the same tetramer. Unless otherwise noted, all antibumping restraints were turned off for the ions and waters in the selectivity filter. The same reflections were omitted for the calculations of $R_{free}$ in order to avoid model bias in this statistic. Attempts to identify heavy ion positions were made both by inspection of anomalous difference maps directly and with Phaser (McCoy *et al*, 2007) using the known partial structures in the phase calculations.

## Electrophysiology

HEK293T cells were transiently transfected with a mammalian cell expression plasmid, pTracer CMV2, containing an N-terminally His-tagged NavMs gene, seeded onto glass coverslips, and placed in a perfusion chamber where extracellular conditions could be altered, as previously described (Bagnéris *et al*, 2014). All cells were voltage-clamped in the whole-cell configuration. Unless otherwise noted, the intracellular (pipette) solution contained (in mM) NMDG (90), NaCl (30), HEPES (10), EGTA (5), CaCl$_2$ (0.5) and pH was adjusted to 7.4 with HCl. When testing the relative permeability of monovalent cations, the bath solution contained (in mM) X-Cl (150), HEPES (10), EGTA (5), CaCl$_2$ (0.5) and the pH was adjusted with X-OH, where the X is the indicated monovalent cation. When testing the relative permeabilities of divalent cations, the bath solution contained X-Cl$_2$ (110), HEPES (10), EGTA (5), CaCl$_2$ (0.5) and the pH was adjusted with X(OH)$_2$, where the X is the indicated divalent cation. The bath solutions used for Appendix Fig S1 were the same as above, but the pipette solution was replaced with NMDG (120), EGTA (5), HEPES (10), CaCl$_2$ (0.5) and the pH adjusted to 7.4 with HCl. For the cells where monovalent ion current was measured (Appendix Fig S1C), 5 mM of the monovalent cations was added to the internal saline solution. When testing the permeability of divalent cations, 2 mM XCl$_2$ was added to the internal saline (buffered by 5 mM EGTA: [Free-Ba$^{2+}$] = 290 μM; [Free-Mg$^{2+}$] = 75 μM and [Free-Ca$^{2+}$] = 105 nM). Liquid junction potentials were measured using a salt bridge as described by Neher (1992) and were comparable to values calculated using Clampex software (Axon Instruments, Inc.). Data were analysed by Igor Pro 6.00 (Wavemetrics, Lake Oswego, OR). Residual leak (> −100 pA) and capacitance were subtracted using a standard P/4 protocol. Current–voltage relationships were fit with (V-V$_{rev}$)/{1 + exp[(V-V$_{1/2}$)/k]}, where V$_{rev}$ is the extrapolated reversal potential. $E_{rev}$ was used to determine the relative permeability of monovalent cation X to Na$^+$ (P$_x$/P$_{Na}$) according to the following equation:

$$\frac{P_x}{P_{Na}} = \frac{\alpha_{Nae}}{\alpha_{xe}}\left[\exp\left(\frac{\Delta E_{rev}}{RT/F}\right)\right]$$

where $\Delta E_{rev}$, $\alpha_x$, $R$, $T$ and $F$ are the reversal potential, effective activity coefficients for cation $x$ ($i$, internal and $e$, external), the universal gas constant, absolute temperature and the Faraday constant, respectively. For these pseudo-bionic conditions, we assumed that the internal NMDG is impermeant. The effective activity coefficients ($\alpha_x$) were calculated using the following equation:

$$\alpha_x = \gamma_x[X]$$

where $\gamma_x$ is the activity coefficient and $[X]$ is the concentration of the ion. For calculations of membrane permeability, activity coefficients ($\gamma$) were calculated using the Debye–Hückel equation: 0.74, 0.76, 0.72, 0.71, 0.69, 0.35, 0.29, 0.27 and 0.27 correspond to Na$^+$, Li$^+$, K$^+$, Rb$^+$, Cs$^+$, Mg$^{2+}$, Ca$^{2+}$, Sr$^{2+}$ and Ba$^{2+}$, respectively. To determine the relative permeability of divalent cations to Na$^+$, the following equation was used:

$$\frac{P_x}{P_{Na}} = \frac{\left\{\alpha_{Nai}\left[\exp\left(\frac{E_{rev}F}{RT}\right)\right]\left[\exp\left(\frac{E_{rev}F}{RT}\right)+1\right]\right\}}{4\alpha_{xe}}$$

$E_{rev}$ for each cation condition was corrected to the measured liquid junction potentials (−4.4 to 3.4 mV). In some cationic conditions,

no inward (negative) voltage-dependent currents could be activated, but $E_{rev}$ was measured as $\leq -4$ mV. In these cases, the lower limit of $P_x/P_{Na}$ was reached (0.1) due to low levels of endogenous $Cl^-$ and nonselective currents in HEK293T cells affecting apparent $E_{rev}$.

For the experiments in Appendix Fig S10, solution delivery was achieved by a stepper mechanism that selects which tube is directed at the patched cell. NavMs $I_{Na}$ often progressively decreases over time by $\geq 3\%$/min (termed "rundown") using this method. This effect was estimated by fitting the decay of $I_{Na}$ during the control perfusion to a linear equation that was extrapolated over the time course of the experiment. Block of $I_{Na}$ by transition metals (Appendix Fig S10) was calculated by subtracting the control $I_{Na}$ rundown regression from $I_{Na}$ after 1.5 min of extracellular application of $Ag^+$ or $Cd^{2+}$. Percent $I_{Na}$ block was calculated by $[(I_{metal} - I_{control})/I_{control}] \times 100$, where $I_{control}$ is the extrapolated rundown current and $I_{metal}$ is the amount of current 1–1.5 min after transition metal application. Transition metal stocks were initially formulated in DMSO and diluted 300–1,000× into extracellular saline solutions. For Appendix Fig S10, the extracellular saline contained (mM) (100) NaCl, (10) HEPES and (X) AgNO$_3$ or CdCl$_2$ (at concentrations indicated in the figure), where all saline conditions were osmotically balanced to 300 ($\pm$ 5) mOsm with mannitol.

The solutions used for inside-out single-channel measurements (Appendix Fig S8) were the same as the standard recording whole-cell recording condition, except that the pipette and bath saline were switched. After capturing the single-channel events by the mutant E178D in control conditions, the membrane potential was held at 0 mV and 500 μM QX-314 was exchanged into the bath for approximately 1 min.

### Electrostatics calculations

Poisson–Boltzmann calculations were carried out using the APBSmem program (Callenberg *et al*, 2010) in the manner previously employed for the proton-gated urea channel (Strugatsky *et al*, 2013) and the CorA channel (Dalmas *et al*, 2014). Modelling of channels with mutant SF sequences was based on the PDBID 5BZB coordinates, with the sequence altered for calculations of the E178D mutant using Modeller (Eswar *et al*, 2006). The program PDB2PQR (Dolinsky *et al*, 2004, 2007) was used to add hydrogen atoms and assign atomic charges and radii to each generated configuration. Partial atomic charges and radii of the protein were taken from the CHARMM parameter sets (Sitkoff *et al*, 1994). The NavMs pore was oriented with its symmetry axis coinciding with the *z*-axis. A $300 \times 300 \times 300$ Å$^3$ map with grid points of $97 \times 97 \times 97$ was employed. The implicit membrane hydrophobic slab was defined between $z = 15$ and $-25$ Å. This corresponds to a hydrophobic slab thickness of 40 Å. The dielectric constants used for protein, membrane and water were 2, 2 and 80, respectively. The ionic strength was 100 mM with coulomb charges $+ 1$ and $-1$, and radius 2.0 Å. The water probe radius was 1.4 Å. The electrostatic energy was calculated for adding an ion at the pore centre at 3 Å intervals along the entire pore length. A range of radii was used for both monovalent and divalent ions to assess the effect that changing ionic radius had on the electrostatic potential energy experienced by an ion. The ionic radii discussed in the results and shown in figures

correspond to the Born radii for the ions concerned: Na$^+$ 1.68 Å, K$^+$ 2.2 Å, Li$^+$ 1.39 Å, Ca$^{2+}$ 1.73 Å.

### Data deposition

Atomic coordinates and structure factors for the wild-type and mutant proteins have been deposited in the Protein Data Bank under accession codes PDBID 5BZB and 4X88, respectively.

**Expanded View** for this article is available online.

### Acknowledgements

We thank Dr. Ambrose Cole (Birkbeck College) for help with crystallographic data collection and the beamline scientists at the Diamond Light Source, UK, Soleil, France, and the ESRF, France. This work was supported by grants (BB/H01070X, BB/L006790 and BB/L02625) from the UK Biotechnology and Biological Science Research Council (BBSRC) to BAW. CEN was supported, in part, by a grant from Pfizer Neusentis (to BAW). PGD was supported by National Institutes of Health Grant T32-HL007572 and NIH Pathway to Independence (PI) Award (K99/R00) from the National Institute of Diabetes and Digestive and Kidney Diseases (NIDDK). The crystallographic research leading to these results has received funding from the European Community's Seventh Framework Programme (FP7/2007–2013) under BioStruct-X (grant agreement N°283570).

### Author contributions

CB performed the molecular biology, and she and ASu purified the proteins, produced and screened the crystals. CEN, CB and ASu undertook the crystallographic data collection, structure solution and refinements. CEN performed the electrostatic calculations. CB, CEN, ASu and BAW analysed the structures. PGD made the constructs and he and ASc conducted the electrophysiological studies; PGD and DEC analysed the electrophysiology results. BAW wrote the initial draft of the manuscript, which was discussed, modified and approved by all authors. BAW and DEC supervised the project.

### Conflict of interest
The authors declare that they have no conflict of interest.

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
