## [Review Process File · The EMBO Journal]

Manuscript EMBO-2015-93285

Molecular Basis of Ion Permeability in a Voltage-Gated Sodium Channel

Claire E. Naylor, Claire Bagn ris, Paul G. DeCaen, Altin Sula, Antonella Scaglione, David E. Clapham and B.A. Wallace

Corresponding author: B.A. Wallace, Birkbeck College, University of London

Review timeline:

Submission date:	15 October 2015
Editorial Decision:	30 November 2015
Revision received:	17 December 2015
Editorial Decision:	15 January 2016
Accepted:	18 January 2016

Transaction Report:

Editor: Daniel Klimmeck

1st Editorial Decision

30 November 2015

Thank you for the submission of your manuscript entitled 'Molecular Basis of Ion Permeability in a Voltage-Gated Sodium Channel' (EMBOJ-2015-93285) to The EMBO Journal. Your study has been sent to four referees, and we have received reports from all of them, which I copy below.

As you will see, the referees acknowledge the potential high interest and novelty of your work, although they also express major technical concerns that would need to be addressed before they can support publication of your manuscript in The EMBO Journal. In particular, referee #2 points out the need for you to improve the resolution and statistical values of both the structures and the supporting electron density maps in order to distinguish between sodium ions and water molecules (see pt. 1). This view was also reflected in the comments from referees #3 and #4 (pts 1 and 2). Another important point raised is the question whether the selectivity filter mutant has structural consequences (see ref #3) and further mutational analysis could provide complementary mechanistic insights (see ref #2 pt. 2). I judge the comments of the referees to be generally reasonable, thus we are in principle happy to invite you to revise your manuscript experimentally to address the referees' comments.

While I realize that solving a new structure at higher resolution would probably not be feasible within the timeline of a revision, I do agree that the study would strongly benefit from an improved

resolution. Thus, I would like to ask your input on whether it was in your view possible to address these referees' points and improve the resolution by e.g. integrating various data sets. In light of these structural comments, I would send a revised manuscript back to referees #2 and 4 for re-review. We appreciate that the required revision may well be beyond the scope of the current project and we would therefore understand if you chose to instead consider publication elsewhere.

In any case, please contact me if you have any questions, need further input on the referee comments or if you anticipate any problems.

Please be aware that it is 'The EMBO Journal' policy to allow a single round of revision only and that, therefore, acceptance of the manuscript will essentially depend on the completeness of your responses included in the next version of the manuscript.

We generally allow three months as standard revision time in the first instance. As a matter of policy, competing manuscripts published during this period will not be taken into consideration in our assessment of the novelty presented by your study ("scooping" protection). Nevertheless, please contact me as soon as possible upon publication of any related work in order to discuss how to proceed. Should you foresee a problem in meeting this three-month deadline, please let us know in advance and we may be able to grant an extension.

When preparing your letter of response to the referees' comments, bear in mind that this will form part of the Review Process File, and will therefore be available online to the community. For more details on our Transparent Editorial Process initiative, please visit our website:

http://emboj.msubmit.net/html/emboj_author_instructions.html#a2.12

As you have probably seen already, every paper now includes a 'Synopsis', displayed on the html and freely accessible to all readers. The synopsis includes a 'model' figure as well as 2-5 one-sentence bullet points that summarize the article. I would appreciate if you could provide this figure and the bullet points.

Finally, in order to ensure good reporting standards and to improve the reproducibility of published results, our guidelines to authors are consistent with the Principles and Guidelines for Reporting Preclinical Research issued by the NIH in 2014. Accordingly, we now require the submission of a completed author checklist, which covers in a systematic manner your practices regarding animal welfare, human subjects, data deposition, statistics and research ethics. It needs to be filled (most of the fields will not apply to your study in particular) and returned to the editorial office at revision, either via the online submission system as a supplementary file or by email (contact@embojournal.org). Please, click on the link below and follow the instructions to download the checklist file:

<http://emboj.embopress.org/authorguide>

Again, please contact me at any time during revision if you need any help or have further questions.

Thank you for the opportunity to consider your work for publication. I look forward to your revision.

REFeree REPORTS:

Referee #1:

Naylor et al. provide the first "picture" of Na⁺ ions trapped in the pore of a voltage-gated Na⁺ channel. Voltage-gated Na⁺ channels are one of the most important classes of ion channels, but how the channels achieve ionic selectivity and permeation is much less understood than K⁺ channels and

Ca²⁺ channels. The data in this paper will likely change this. The authors achieved a high-resolution (2.7 Å) structure of the ion selectivity pore of NavMs, a bacterial voltage-gated Na⁺ channel. The structure of the Na⁺-soaked crystal shows three Na⁺ ions in an open pore. It also shows that Na⁺ ions in the pore can be at least partially hydrated. In addition, the ion-binding sites in the Na⁺ channel are formed by the side chains, unlike the K⁺ channel binding sites formed by the polypeptide backbone carbonyl groups. This use of side chains is perhaps not surprising given previous data from Ca²⁺ channels, but it is gratifying to see the formation of the binding sites. To confirm the importance of the Na⁺ binding sites in channel conductance, the authors mutated one of the sites (E178D) and demonstrated with crystal structure a loss of a Na⁺ occupation, and with patch clamp recording an ~ 50% reduction in single channel conductance.

The studies are well done with a nice combination of structural, electrophysiological and computational studies. They help our understanding of ionic selectivity and permeation in voltage-gated Na⁺ channels. The paper is well written and suitable for publication. I have only three minor comments.

1. The E178 site is important for both selectivity and permeation. The electrostatic calculation predicts the altered selectivity in the E178D mutant. Would it be possible to speculate how the mutation leads to the reduction in single channel conductance?
2. Fig. 1A, B, the Y-axes of I-V curves lack units (pA/pF?)
3. Fig. S3, is there major difference in single channel opening probabilities between the WT and the E178D mutant?

Referee #2:

EMBOJ-2015-93285

Comments for the authors:

First, this author think it is not necessary to say that voltage-gated sodium channel is physiologically important. One of the distinct features of voltage-gated sodium channel is its high ion selectivity, which is mainly defined by the structure and sequence of the selectivity filter. Several crystal structures of prokaryotic voltage-gated sodium channel (NavAb, NavMs, NavRh, NavAe) and Ca²⁺-selective NavAb mutant (named CavAb) have been reported, and some models has been proposed based on the results of MD simulations and biochemical analyses. However, the precise number, sites, and forms (i.e. hydrated and/or dehydrated) of bound sodium ions have remained elusive, because of the lack of high-resolution structure of the voltage-gated sodium channel with bound sodium ions.

In this manuscript, Naylor et al. report the crystal structures of prokaryotic voltage-gated sodium channel, NavMs in sodium-binding form. By improving the maximum resolution from 3.5 Å to 2.7 Å, the authors clarified three sodium-binding sites and provide the direct evidence of sodium recognition mechanism mediated by the ordered and partially ordered water molecules in the selectivity filter. The authors also presented the detailed ion selectivity of NavMs based on the electrophysiological experiments using HEK cells and electrostatic potential calculations. Combining these structural, functional and computational analyses on both wild type and selectivity filter mutants, the authors revealed that the recognition by the side chain of Glu178 (composing the most extracellular binding site (i.e. NaI site)) is particularly important for its high sodium selectivity. Furthermore, by comparing to the structure of voltage-gated potassium channel KcsA that recognizes "dehydrated" potassium ions by "backbone only", the authors provided the structural evidence that the "side chain" of the selectivity filter residues and "hydrated" forms of sodium ions are critical for sodium recognition and permeation by NavMs.

Although this is an interesting article that would enhance the understanding of the ion permeation mechanism of voltage-gated sodium channel, it contains several major concerns, including the most

severe ones on the statistical values of X-ray diffraction data and the quality of the electron density maps. Because all the author's discussion is based on the electron density map at the resolution of 2.7 \AA , these points are extremely important and should be scrutinized carefully before the publication.

Major concerns

1. Resolution and density map

The authors discussed about the sodium ions and water molecules in the selectivity filter, on all of which is based on the electron density maps. Therefore, it is more important than anything else to ensure the quality and reliability of the density maps. In order to discuss the "precise" sodium recognition mechanism including sodium-binding sites and coordinated water molecules, this reviewer strongly requests the higher-resolution structures, or, at least, the improvement of the density maps. If these were not possible, the novelty of this article would be reduced because the sodium-binding sites have already been predicted roughly and also it has been predicted that some of the sodium ions in the selectivity filter are recognized in hydrated forms. Here are the major concerns on the resolution and the density maps;

- i. Although the authors provide additional discussion in the Appendix, in this reviewer's opinion, the resolution of 2.7 \AA itself is not sufficient (or on the border line, at best) to discuss the location and coordinated state of substrate ions. The densities of water molecules are quite poor, and the density maps of even the selectivity filter residues seem to be poor (Fig. 2A) although the statistical highest resolution is 2.7 \AA . The best way is, eventually, to collect the better data set and solve the sodium-bound structure from the single crystal in higher resolution. Since, however, it is very difficult, as usual, the alternative way is to merge the multiple partial data sets. Because the space group of both wild type and E178D mutant (C2221) has high symmetry, it seems to be realistically possible to collect the several partial higher-resolution data.
- ii. Is there any problem to extend the highest resolution? Although the I/I_0 and $CC1/2$ value is quite high ($I/I_0 = 3.2$ and $CC1/2 = 85.3$ at the highest shell) and other values (B-factor, $R_{\text{work}}/R_{\text{free}}$, Ramachandran plots, completeness, redundancy, etc...) are also still good, the resolution cutoff point is too low. If the statistic values such as I/I_0 and $CC1/2$ are significantly high enough and the density maps (especially of the coordinated water molecules) are improved by using the higher resolution data, the higher resolution structure would be definitely more useful for the context of this article. The authors should reprocess the data and replace structures to higher-resolution ones, if possible.
- iii. In the Appendix table S1 (Data collection and refinement statistics), it is somewhat strange that although the I/I_0 value of the highest shell is quite high (3.2), the $CC1/2$ of the same shell is decreased to 85.3%. In general, the $CC1/2$ value of the shell with such a high I/I_0 would show almost 100% correlation. In the data processing, "sigma cut off" operation or any additional operation has been done?
- iv. In the Fig. 2A, the $2F_o - F_c$ map of the selectivity filter residues are depicted contoured in 2 but the electron density map contoured in other values such as 1 and 1.5 should be shown as an appendix figure in order to ensure the reliability of the presented resolution.

2. Additional selectivity filter mutants should be analyzed

In the case of NavAb, the ion selectivity has been altered dramatically when triple mutations were introduced into the selectivity filter (changed from LESWSM to LDDWSD). The authors investigate several mutants including E178 mutants, but neither S179 nor M182 mutant. These two residues might change the ion selectivity dramatically. Therefore, additional three selectivity filter mutants of LEDWSM LDDWSM LESWSD should be analyzed by the whole cell patch clamp experiment

3. Permeability of Mg^{2+}

By using the electrophysiological experiments and electrostatic potential calculations, the authors examine the ion selectivity of NavMs for various mono/divalent-cations, but not for magnesium ion that is the most abundant divalent cation in the cells. Although magnesium ion is divalent, it has much smaller ion radii than calcium and sodium ions, and therefore there is a possibility that NavMs permeates magnesium ion. In order to understand the detailed ion selectivity of NavMs, the electrophysiological analyses and electrostatic potential calculations should be done on magnesium ion as well as other mono/divalent cations.

Minor concerns

1. Page 6 Line 8-9

The authors say that "... sodium ion sites are seen in the centres of the selectivity filter...". However the density of the NaI site seems to extend more laterally than those of NaII and NaIII sites. NaI site is corresponding to the S1 site predicted by the MD simulation of NavAb (Ulmschneider et al., PNAS (2013)). According to this paper, S1 site is the most dominant site and asymmetrically disposed along the pore direction, due to the direct interaction with the carbonyl group of Glu residue. This is consistent with the extension of the density. Therefore, this reviewer guess the sodium ion of NaI site is not permeated in the centre of the selectivity filter.

2. The improvement of the resolution (from $3.5 \approx$ to $2.7 \approx$) is an excellent point of this report. Purification and crystallization procedures are, however, almost the same as those of the previous report. Although the luck or chance is often a big factor for crystallization, the authors should mention about how they improve the data, if any strategies were used or if anything could be learnt from the large number of crystallization trials.

3. The paper on the crystal structure of Nav from a bacterium *Alkalilimnicola ehrlichei* should be cited. (Shaya et al., 2014 (J. Mol. Biol.) doi:10.1016/j.jmb.2013.10.010)

4. The authors refer to the S1, S2, S3 and S4 site in the manuscript, but not to the S0 site. Is there any density observed in the place near the S0?

5. Anomalous peaks at the vestibule of the channel derived from co-crystallized/soaked metals and thalium should be described as a supplemental figure.

6. Please add the names of the residues on Fig. 2A or 2B.

7. In the "Electrophysiology" section of Materials and Methods, the resolutions of the images of the equations are too low to read. Please replace them to the higher resolution ones.

8. Page 32 Table 1A & 1B

The notation of "plus-minus" should be drawn with the symbolic font (like \pm), not with the plus symbol (+) with under bar.

9. Page 3 Line 7

each a structural domain -> each of which is composing a structural domain

10. Page 7 Line 12

"..., while NaI and NaII are $0.2 \approx$ are further apart ..." should be change to "..., while NaI and NaII are $0.2 \approx$ further apart ..."

11. Page 7 Line 13

"Thus sodium ions thus appear to be ..."

The word "thus" is a duplicate in this sentence.

12. Page 10 Line 14

"...D178 is replaced with the shorter side chain in the mutant..." must be change to "...E178 is replaced with the shorter side chain in the mutant..."

13. Page 11 Line 21

...very similar... -> ...similar...

(Because NaII site is not so similar to each other.)

Referee #3:

The ms of Naylor et al describes a crystal structure of the bacterial sodium channel NavMs. Previously, crystal structures of several prokaryotic sodium channels have been solved and have been related to different functional states of the channel. Hitherto resolution of crystal structures of the NavMs channel were at 3.5 \AA , too low to resolve sodium ion locations within the selectivity filter. Here Naylor et al present a structure obtained from crystals diffracting to a resolution of 2.7 \AA . Crystals were grown in the presence of sodium. Attempts to substitute sodium by other cations were unsuccessful, suggesting that the selectivity filter contained tightly bound sodium ions carried through the purification steps. This situation makes it difficult to assign electron density within the SF as arising from sodium ions. However, as discussed in the appendix, it is very challenging at this (still relatively low) resolution to unambiguously distinguish between sodium ions and water molecules. Both have 10 electrons. Though a discrete spherical density can be seen at a high sigma cutoff (Appendix Figure S6), it is only an indication, but still no proof for defined locations of sodium ions within the selectivity filter. Therefore, the assignments of sodium binding sites are still very speculative. Furthermore, Naylor et al claim that they obtained a sodium channel pore structure in an opened state. It is unclear, which opened state Naylor et al refer to. Is it an opened conformation of the pore with activation gate and inactivation gate opened or one with activation gate opened, but inactivation gate closed or activation gate closed and inactivation gate opened? The extensive work on potassium channel structures indicates that it is very difficult, if not impossible, to define at this low resolution a genuine structure of an 'opened' channel, especially when a membraneous environment and a defined membrane potential are absent. Furthermore, Naylor et al claim that sodium ions are capable of binding to several closely-related sites as they pass through the selectivity filter 'possibly due to the presence of intervening water molecules along the length of the selectivity filter'. This claim lacks sufficient evidence. Apart from a higher structural resolution a molecular dynamics study of the passage of sodium ions through the selectivity filter would be useful.

In the second part of their ms Naylor et al study the effect of mutating pore residue E178 to aspartic acid (E178D). The mutation appears to alter the first sodium binding site (NaI) and has dramatic effects on sodium channel properties including single channel conductance, selectivity, current density. This may indicate considerable structural differences between wild type and mutant channel. For example, the mutation may affect the network behind the selectivity filter that stabilizes the pore structure. It is important to provide this information. This kind of mutation can have dramatic effects for potassium channel pore structures.

The study is potentially very interesting to a wider audience as it discusses an important aspect of sodium ion transport across the plasma membrane. However, the results shown provide insufficient evidence for the claims made. Probably for this reason the results section contains many very speculative statements, which should be moved to Discussion.

Minor (or major?) point: Who is Antonella. Please, give the full name. Also, indicate whose present address is at the institute of Molecular Biology and Pathology in Rome.

Referee #4:

Naylor et al report the structure of the ion-selective pore of a bacterial sodium channel with bound sodium ions. Bacterial sodium channels are important model systems for the eukaryotic channels, for which there are no high-resolution structures. A number of structures of bacterial sodium channels have been reported, not least by the Birkbeck group, but none of them have revealed the position of the bound sodium ions in the channel, which is important for understanding how it works and how ions are selected. The comparison to the ion-bound selectivity filter of potassium channels is particularly instructive, as it shows that potassium ions are desolvated for passage through the channel, whereas sodium ions are not. This appears to be a sufficient, simple explanation of ion selectivity. The observation that the E178D mutant has two instead of three sodium ions is gratifying and reassuring.

This is an interesting, succinct and timely manuscript. Nevertheless there are a few points that merit further attention.

1. One query relates to the electron density in the channel and its interpretation as sodium ions or water molecules. As the authors point out, these are difficult to tell apart in an x-ray structure at this resolution. Yet whether the channel contains three or four sodium ions is an important point. The densities in the channel are surprisingly low, as they are contoured at 2.5 or even 2.2 sigma. This suggests that the occupancy is low - at full occupancy, sodiums should show up at 5 sigma or higher. What could be the reason for the weak density and low sodium occupancy?

2. The density assigned to a water molecule in the S3 position is even weaker, which must make it hard to be sure it is not a low-occupancy sodium ion, especially since it seems to be coordinated by four water molecules with the same geometry as the sodium ion in S4, as far as I can tell from Figure 2A. Have the occupancies of the sodium ions and water molecules been refined? What are they? Omit maps for the sodiums and the supposed water molecule on the channel axis should be shown. Unless the authors can rule out that there are actually four sodium ions bound, this should at least be discussed as a possibility, to be on the safe side. It will not take anything away from the significance of the paper.

3. Are there any conformational changes in the selectivity filter with and without bound sodium? What are they?

4. The authors state that dehydrated potassium ions are of about the right size to pass through the channel, but that this would be energetically unfavourable. How about hydrated potassium ions? Are they too large, and by how much? A table of ionic radii of hydrated and dehydrated sodium, potassium and calcium ions (with error margins!) would be instructive.

5. It would be interesting to know why the crystals now diffract better than before. Is it possible to say what made the difference?

Minor points:

6. p5, top para: "... suggesting that protein-ion interactions...". 2nd para: "Thus sodium ions appear..." - check for similar mistakes.

7. The name of one of the authors is incomplete.

1st Revision - authors' response

17 December 2015

Comments for the Referees: (*Note: referees comments are in italics, our responses are in bold. Points from the referees comments we would like to emphasise are in bold italics; additions to the manuscript are indicated in italics.*)

Referee #1:

*Naylor et al. provide the first "picture" of Na⁺ ions trapped in the pore of a voltage-gated Na⁺ channel. Voltage-gated Na⁺ channels are one of the most important classes of ion channels, but how the channels achieve ionic selectivity and permeation is much less understood than K⁺ channels and Ca²⁺ channels. **The data in this paper will likely change this.** The authors achieved a high-resolution (2.7 Å) structure of the ion selectivity pore of NaVMs, a bacterial voltage-gated Na⁺ channel. The structure of the Na⁺-soaked crystal shows three Na⁺ ions in an open pore. It also shows that Na⁺ ions in the pore can be at least partially hydrated. In addition, the ion-binding sites in the Na⁺ channel are formed by the side chains, unlike the K⁺ channel binding sites formed by the polypeptide backbone carbonyl groups. This use of side chains is perhaps not surprising given previous data from Ca²⁺ channels, but it is gratifying to see the formation of the binding sites. To confirm the importance of the Na⁺ binding sites in channel conductance, the*

authors mutated one of the sites (E178D) and demonstrated with crystal structure a loss of a Na⁺ occupation, and with patch clamp recording an ~ 50% reduction in single channel conductance.
No response required.

The studies are well done with a nice combination of structural, electrophysiological and computational studies. They help our understanding of ionic selectivity and permeation in voltage-gated Na⁺ channels. The paper is well written and suitable for publication. I have only three minor comments.

No response required.

1. The E178 site is important for both selectivity and permeation. The electrostatic calculation predicts the altered selectivity in the E178D mutant. Would it be possible to speculate how the mutation leads to the reduction in single channel conductance?

The structure demonstrates that there is one less binding site in the mutant channel than in the wild-type structure, which could be consistent with few ions passing through the channel per unit of time, and therefore a lower current density.

A sentence to this effect is now added at the end of the section on the mutant.

2. Fig. 1A, B, the Y-axes of I-V curves lack units (pA/pF?)

The peak currents from Li⁺, Cs⁺, K⁺, Rb⁺ and Ba²⁺ conditions were divided by the peak current measured from in sodium condition for each cell. To clarify, we have added this statement to the figure legend: "...where current magnitudes were normalized to the maximum inward current measured in the sodium condition." Since this normalization does not have units, the figure legend remains without units.

3. Fig. S3, is there major difference in single channel opening probabilities between the WT and the E178D mutant?

This is a good point. We performed the analysis but observed no difference in the open probability between the two channels. Below is a table listing the P_o measured from the first 80 ms after depolarization (N=3, Error = standard deviation, Measured for patch in which only one channel was present). We have not added this to the manuscript (but could add it as an additional table in the appendix if the editor deems appropriate) but added a comment about this to the text of the manuscript indicating WT and mutant single channel conductances are not significantly different.

V _m (mV)	Wt	E178D
-80	0.57 (± 0.10)	0.62 (± 0.15)
-60	0.74 (± 0.11)	0.78 (± 0.12)
-40	0.6 (± 0.09)	0.62 (± 0.07)
-20	0.49 (± 0.04)	0.53 (± 0.06)
0	0.46 (± 0.04)	0.39 (± 0.09)
10	0.28 (± 0.04)	0.23 (± 0.08)
20	0.21 (± 0.05)	0.19 (± 0.06)

Referee #2:

First, this author think it is not necessary to say that voltage-gated sodium channel is physiologically important.

We have added two sentences to the beginning of the introduction to emphasise the physiological importance of these channels.

One of the distinct features of voltage-gated sodium channel is its high ion selectivity, which is mainly defined by the structure and sequence of the selectivity filter. Several crystal structures of prokaryotic voltage-gated sodium channel (NavAb, NavMs, NavRh, NavAe) and Ca²⁺-selective NavAb mutant (named CavAb) have been reported, and some models has been proposed based on the results of MD simulations and biochemical analyses. However, the precise number, sites, and forms (i.e. hydrated and/or dehydrated) of bound sodium ions have remained elusive, because of the lack of high-resolution structure of the voltage-gated sodium channel with bound sodium ions.

No response required.

In this manuscript, Naylor et al. report the crystal structures of prokaryotic voltage-gated sodium channel, NavMs in sodium-binding form. By improving the maximum resolution from 3.5 Å to 2.7 Å; **the authors clarified three sodium-binding sites and provide the direct evidence of sodium recognition mechanism mediated by the ordered and partially ordered water molecules in the selectivity filter.** The authors also presented the detailed ion selectivity of NavMs based on the electrophysiological experiments using HEK cells and electrostatic potential calculations. Combining these structural, functional and computational analyses on both wild type and selectivity filter mutants, the authors revealed that the recognition by the side chain of Glu178 (composing the most extracellular binding site (i.e. NaI site)) is particularly important for its high sodium selectivity. Furthermore, by comparing to the structure of voltage-gated potassium channel KcsA that recognizes "dehydrated" potassium ions by "backbone only", the authors provided the structural evidence that the "sidechain" of the selectivity filter residues and "hydrated" forms of sodium ions are critical for sodium recognition and permeation by NavMs.

No response required.

Major concerns

1. Resolution and density map

The authors discussed about the sodium ions and water molecules in the selectivity filter, on all of which is based on the electron density maps. Therefore, it is more important than anything else to ensure the quality and reliability of the density maps. In order to discuss the "precise" sodium recognition mechanism including sodium-binding sites and coordinated water molecules, this reviewer strongly requests the higher-resolution structures, or, at least, the improvement of the density maps. If these were not possible, the novelty of this article would be reduced because the sodium-binding sites have already been predicted roughly and also it has been predicted that some of the sodium ions in the selectivity filter are recognized in hydrated forms.

The referee is correct that the sodium binding sites have already been predicted "roughly", but the importance of this work is that it is the first time they have been defined experimentally. Predictions make many assumptions and without experimental verification or definition, they remain hypothetical. This paper shows the ion locations, the extent of their interactions with the polypeptide and, importantly, that there are no "direct" contacts with the side chains or backbone, and hence that the interactions involve intermediary water molecules. These are major findings, and ones that can be clearly defined at the resolution of the current maps. This study will provide an important reference point for sodium channel structures, and a fiduciary point for further modelling, calculation, and dynamics studies, in the way the first studies of potassium channels showed that the potassium ions were dehydrated and interacted with the polypeptide backbone, work that has been extensively cited and figures that have been frequently reproduced.

Here are the major concerns on the resolution and the density maps;

i. Although the authors provide additional discussion in the Appendix, in this reviewer's opinion, the resolution of 2.7 Å; itself is not sufficient (or on the border line, at best) to discuss the location and coordinated state of substrate ions. The densities of water molecules are quite poor, and the density maps of even the selectivity filter residues seem to be poor (Fig. 2A) although the statistical highest resolution is 2.7 Å;. The best way is, eventually, to collect the better data set and solve the sodium-bound structure from the single crystal in higher resolution. The structure in this paper at 2.7 Å is the highest resolution structures yet produced for any sodium channel, and is much higher resolution than the only other structure with ions present in the channel (3.2 Å). They are of high quality and sufficient (beyond the "borderline" that the referee suggests) to identify the contents of the channel selectivity filter. We note that at this resolution water molecules are clearly resolved (and as is normal, have been added to the

model), as are sidechain features and even, in well-resolved regions of the map, carbonyl ‘bumps’ on the mainchain. This resolution is therefore clearly sufficient to observe peaks in the map due to sodium ions (which have the same number of electrons as water). Improving the resolution beyond this would be very challenging (it has taken us 3 1/2 years to improve the resolution beyond that of the first structures (3.5 Å resolution) that showed any density in the selectivity filter (McCusker et al, 2012)). The density in the SF of these new crystals is very clear.

Since, however, it is very difficult, as usual, the alternative way is to merge the multiple partial data sets. Because the space group of both wild type and E178D mutant (C2221) has high symmetry, it seems to be realistically possible to collect the several partial higher-resolution data. It has taken considerable ingenuity to produce the maps of the quality shown, as these crystals (as is common with type II membrane protein crystals) suffer from significant anisotropy and disorder in one dimension. We have collected approximately 150 datasets from different and identical conditions, and these are the highest resolution data obtainable. In addition we have tried many processing and refinement procedures and conditions, and concluded that the cutoffs and methods described in this study produce the most correct and highest resolution maps possible. We do not feel it is likely that either further data collection or alternative processing methods will result in significant improvements in resolution, but believe that the maps presented and the strategies tested for refinement (as in Appendix Figure S2 and the associated discussion) clearly indicate the nature and locations of the ions.

While this referee is correct that sometimes resolution can be improved by collecting and merging multiple datasets, this technique relies on a good degree of isomorphism between crystals. This crystal form lacks extensive contacts in the c-axis and therefore shows considerable variation in this dimension (from 322 to 340 Å). A change in unit cell dimension of $d(\text{min})/4$ is allowable to maintain isomorphism (at 2.7 Å this is approximately 0.6 Å) (Garmen & Murray, 2003); clearly the change in our c-axis far exceeds this, so we are unable to effectively merge crystals, as suggested by the referee (although we did attempt to do so).

ii. Is there any problem to extend the highest resolution?

Yes, as with most polytropic membrane proteins, and ion channels in particular, producing crystals that diffract to higher resolution than 2.7 Å is very challenging (<25% of the unique membrane proteins solved to date diffract to this high a resolution, due to packing disorder, the presence of detergent lipid molecules, etc.)

Although the $I/\sigma(I)$ and $CC1/2$ value is quite high ($I/\sigma(I) = 3.2$ and $CC1/2 = 85.3$ at the highest shell) and other values (B-factor, $R_{\text{work}}/R_{\text{free}}$, Ramachandran plots, completeness, redundancy, etc...) are also still good, the resolution cutoff point is too low.

This is not true. The $CC1/2$ is the indication there is lattice disorder, and hence the appropriateness of cutting off the resolution as we have done at 2.7 Å. Indeed, all the crystal forms of NavMs (and for that matter NavAb and NavRh) that have been published contain a disordered C-terminal domain, as described in Bagneris et al, 2013 (and illustrated in figure 4b of that paper). In these type II crystals there are layers of ordered transmembrane domains in detergent alternating with layers of soluble, disordered C-terminal domains. This results in a layer lattice disorder, which manifests as alternate layers of streaky spots on the diffraction images. The difficulties this imposes on data collection means that many of the measures of data quality are impacted, including, as the referee notes, a lower $CC(1/2)$ than might be expected for a given intensity, as well as higher R factors. Because of the known pathology we have been conservative in our resolution cutoff, to ensure that the data and maps are still of sufficient quality to be confident of our interpretation.

If the statistic values such as $I/\sigma(I)$ and $CC1/2$ are significantly high enough and the density maps (especially of the coordinated water molecules) are improved by using the higher resolution data, the higher resolution structure would be definitely more useful for the context of this article. The authors should reprocess the data and replace structures to higher-resolution ones, if possible.

We had already tested the procedures the referee suggests, but for membrane protein crystals such as these with the lattice disorder, they do not produce any actual improvement.

However, to illustrate the effects for the referee, below we list a table with the data refined at 2.4 Å resolution. It can be seen that although the $R_{\text{work}}/R_{\text{free}}$ are not too badly degraded, and the $I/\sigma(I)$ in the outer shell is still only 1.2, the $CC1/2$ drops to 0.68. Although this still is high enough to be included in refinement, it indicates the challenges of this data, and the reason we have selected a very conservative resolution cut-off. More importantly the map is no better.

Data Collection		Refinement	
Resolution (high) Å	45.72-2.4 (2.49-2.40)	Resolution (high) Å	45.72-2.4 (2.46-2.40)
R _{merge}	0.32 (2.559)	No. reflections	
R _{pim}	0.09 (0.713)	R _{work} /R _{free}	19.7/22.2 (23.4/24.9)
CC _{1/2} (%)	0.99 (0.68)	RMSD – Bond lengths	0.007
I/σ	8.2 (1.2)	RMSD – Bond angles	1.105
Completeness (%)	100.0 (100.0)	Ramachandran - favoured	96.91%
Redundancy	13/5 (13.7)	Ramachandran - outliers	0.0%

2F_o-F_c maps shown as grey mesh at 1.0 (light), 1.5 (mid) and 2.0 (dark) rmsd for data re-processed and extended to 2.4 Å.

We have included the above in response to the referee's comment, but not included it in the revised version of the paper, because it is not appropriate to push the data in this way due to the pathology of the crystals. However, we do now note in the materials and methods section that the data was collected to 2.4 Å resolution, but that we have conservatively refined it only to 2.7 Å based on the lattice layer disorder.

Nevertheless, the electron density maps produced at 2.7 Å enable the level of interpretation that we have described, and even if we were to use the higher resolution data (as above) it does not contradict the current interpretations.

iii. In the Appendix table S1 (Data collection and refinement statistics), it is somewhat strange that although the I/σI value of the highest shell is quite high (3.2), the CC1/2 of the same shell is decreased to 85.3%. In general, the CC1/2 value of the shell with such a high I/σI would show almost 100% correlation. This is the result of lattice disorder, a very common feature for membrane protein crystallography, as described above. As described in response to (ii) above, the layer lattice disorder increases the error in our intensity measurements for any given intensity, and therefore the CC(1/2) is lower than might be expected at this intensity.

In the data processing, "sigma cut off" operation or any additional operation has been done? No I/σI cut-off has been applied. To change the cut off or processing as described by the referee would be entirely inappropriate for this type of data.

iv. In the Fig. 2A, the 2Fo-Fc map of the selectivity filter residues are depicted contoured in 2 σ; but the electron density map contoured in other values such as 1 σ; and 1.5 σ; should be shown as an appendix figure in order to ensure the reliability of the presented resolution.

These have been added as an additional Appendix Figure (S8), as requested, and confirm the suitability of the interpretation at this level of resolution.

2. Additional selectivity filter mutants should be analyzed In the case of NavAb, the ion selectivity has been altered dramatically when triple mutations were introduced into the selectivity filter (changed from LESWSM to LDDWSD). The authors investigate several mutants including E178 mutants, but neither S179 nor M182 mutant. These two residues might change the ion selectivity dramatically. Therefore, additional three selectivity filter mutants of LEDWSM; LDDWSM; LESWSD should be analyzed by the whole cell patch clamp experiment;

We did attempt to make a number of mutants in M182 (as previously described in the manuscript) but they did not express in a way that made functional channels. The proposed sites in the CavAb structure did include changes to this residue, but as it is not present at the

contact points for the ion experimentally-determined sites in NavMs, mutants in this site would not be expected to impact on the present conclusions. As the reviewer has noted, the CavAb paper by Payandeh et al, did make numerous changes in order to completely change the selectivity profile of the NavAb channel, but the aim of the mutant in current paper was not to create a channel with a new selectivity but to validate the role of the E178 residue in the existing selectivity pathway.

3. *Permeability of Mg²⁺. By using the electrophysiological experiments and electrostatic potential calculations, the authors examine the ion selectivity of NavMs for various mono/divalent cations, but not for magnesium ion that is the most abundant divalent cation in the cells. Although magnesium ion is divalent, it has much smaller ion radii than calcium and sodium ions, and therefore there is a possibility that NavMs permeates magnesium ion. In order to understand the detailed ion selectivity of NavMs, the electrophysiological analyses and electrostatic potential calculations should be done on magnesium ion as well as other mono/divalent cations.*

Magnesium is unlikely to be a permeant ion due to the size of hydrated Mg²⁺ and the high dehydration energies required to remove waters from Mg²⁺ (4 orders of magnitude lower exchange rate (waters/s) than Na or K: $\Delta H_{\text{hydration}} = -105$ kcal/mol for Na vs -476 kcal/mol for Mg²⁺; Edsall and McKenzie, 1978). Nevertheless, we have calculated the electrostatics for Mg²⁺ (now included in revised Appendix Figure S1, and also mentioned it in the main text). We have also tested Mg²⁺ permeation in whole cell patch clamp and observe no inward voltage dependent currents conducted through the NavMs channel (We estimate the relative $P_{\text{Mg}}/P_{\text{Na}} \leq 0.1$). These data have been added to a revised Appendix Figure S1, which incorporates original Figure 1C and the new data).

Minor concerns

1. *Page 6 Line 8-9. The authors say that "... sodium ion sites are seen in the centres of the selectivity filter...". However the density of the NaI site seems to extend more laterally than those of NaII and NaIII sites. NaI site is corresponding to the S1 site predicted by the MD simulation of NavAb (Ulmschneider et al., PNAS (2013)). According to this paper, S1 site is the most dominant site and asymmetrically disposed along the pore direction, due to the direct interaction with the carbonyl group of Glu residue. This is consistent with the extension of the density. Therefore, this reviewer guess the sodium ion of NaI site is not permeated in the centre of the selectivity filter. The S1 site in the simulation paper is asymmetrically disposed with respect to the SF, but because of the symmetry of the tetramer in the crystal, this would be consistent with the slightly wider (but NOT asymmetric) high level contour electron density seen in the crystal structure, as the referee suggests (and as we have now noted in the results section). The closest interaction is with the Glu residue, but in the symmetric tetramers present in prokaryotic sodium channels, the interaction with all four Glu in the SF should be equivalent. It is not the case as the reviewer suggests, that the ion has not permeated into the centre of the SF. It is important to note that the MD studies are calculations based on a number of approximations, and that the crystal structure in this paper provides actual experimental evidence.*

2. *The improvement of the resolution (from 3.5 Å to 2.7 Å) is an excellent point of this report. Purification and crystallization procedures are, however, almost the same as those of the previous report. Although the luck or chance is often a big factor for crystallization, the authors should mention about how they improve the data, if any strategies were used or if anything could be learnt from the large number of crystallization trials.*

The differences that enabled the improvement in resolution were small but manifold, as is often the case for membrane proteins. The conditions cited in this paper can be compared with those in the 3.5 Å resolution paper, but probably are not appropriate to list in this work. Amongst the changes were: different constructs (full length C-terminal domain vs. fully or partially truncated C-terminal domains), minor changes to the solubilisation and thrombin cleavage procedures, small changes to the salts, imidazole and salt concentrations and pH values, crystal handling procedures, small differences in protein concentration, minor changes to PEG concentrations, etc. No one thing made a large incremental difference. We have now stated this in the paper. These took place successively over a period of 3.5 years! This is the main reason we deem it unrealistic to expect that higher resolution crystals will be easily prepared (as implied by the referee).

3. *The paper on the crystal structure of Nav from a bacterium Alkalilimnicola ehrlichei should be cited. (Shaya et al., 2014 (J. Mol. Biol.) doi:10.1016/j.jmb.2013.10.010)*

This paper was cited and was in the reference list.

4. *The authors refer to the S1, S2, S3 and S4 site in the manuscript, but not to the S0 site. Is there any density observed in the place near the S0?*

There was no density visible in the S0 site, hence it was not discussed. Again it is important to note that the S0 site was a low occupancy site external to the SF, that was proposed based on MD calculations, but the sites discussed in this manuscript are based on experimental evidence. The absence of the S0 site is now discussed in the manuscript.

5. *Anomalous peaks at the vestibule of the channel derived from co-crystallized/soaked metals and thallium should be described as a supplemental figure.*

We attach the anomalous difference maps for the silver- and thallium- soaked crystals below. No significant peaks were observed above background. We don't think this is necessary to include in the supplementary material, but will be happy to do so if the editor so indicates.

Anomalous difference electron density maps for NavMs pore crystals produced in the presence of other cations. In all cases the contour levels (shown as red mesh) were at 3.0σ , and are overlaid on the wildtype sodium crystal structure with the front monomer removed for clarity. Sodium ion sites (from the AB tetramer of the sodium-only containing structure) are shown as blue spheres. A) Crystals grown in the presence of thallium and collected at a wavelength of 0.976 Å. B) Crystals produced after soaking of the wildtype (sodium) crystals with silver nitrate and data collection at a wavelength of 1.77 Å. It is also clear that neither of these anomalous scatterers displace the sodium ions in the selectivity filter.

6. *Please add the names of the residues on Fig. 2A or 2B.*

Done on Fig. 2B, as requested.

7. *In the "Electrophysiology" section of Materials and Methods, the resolutions of the images of the equations are too low to read. Please replace them to the higher resolution ones.*

Corrected in the revised manuscript.

8. *Page 32 Table 1A & 1B*

The notation of "plus-minus" should be drawn with the symbolic font (like \pm), not with the plus symbol (+) with under bar.

Corrected in the revised manuscript.

9. Page 3 Line 7

each a structural domain -> each of which is composing a structural domain

A corrected version is now in the revised manuscript.

10. Page 7 Line 12

"..., while NaI and NaII are 0.2 Å; are further apart ..." should be change to "..., while NaI and NaII are 0.2 Å; further apart ..."

Corrected in the revised manuscript.

11. Page 7 Line 13

"Thus sodium ions thus appear to be ..." The word "thus" is a duplicate in this sentence.

Corrected in the revised manuscript.

12. Page 10 Line 14

"...D178 is replaced with the shorter side chain in the mutant..." must be change to "...E178 is replaced with the shorter side chain in the mutant..."

Corrected in the revised manuscript.

13. Page 11 Line 21

...very similar... -> ...similar... (Because NaII site is not so similar to each other.)

Changed in the revised manuscript as requested by the referee.

Referee #3:

The ms of Naylor et al describes a crystal structure of the bacterial sodium channel NavMs. Previously, crystal structures of several prokaryotic sodium channels have been solved and have been related to different functional states of the channel. Hitherto resolution of crystal structures of the NavMs channel were at 3.5 Å; too low to resolve sodium ion locations within the selectivity filter. Here Naylor et al present a structure obtained from crystals diffracting to a resolution of 2.7 Å. Crystals were grown in the presence of sodium. Attempts to substitute sodium by other cations were unsuccessful, suggesting that the selectivity filter contained tightly bound sodium ions carried through the purification steps. This situation makes it difficult to assign electron density within the SF as arising from sodium ions. However, as discussed in the appendix, it is very challenging at this (still relatively low) resolution to unambiguously distinguish between sodium ions

and water molecules. Both have 10 electrons. Though a discrete spherical density can be seen at a high sigma cutoff (Appendix Figure S6), it is only an indication, but still no proof for defined locations of sodium ions within the selectivity filter. Therefore, the assignments of sodium binding sites are still very speculative. Furthermore, Naylor et al claim that they obtained a sodium channel pore structure in an opened state. It is unclear, which opened state Naylor et al refer to. Is it an opened conformation of the pore with activation gate and inactivation gate opened or one with activation gate opened, but inactivation gate closed or activation gate closed and inactivation gate opened?

The extensive work on potassium channel structures indicates that it is very difficult, if not impossible, to define at this low resolution a genuine structure of an 'opened' channel, especially when a membraneous environment and a defined membrane potential are absent.

We did state it was the activation gate at the intracellular ends of the transmembrane helices, which is what is commonly considered when an "open" state is referred to; but we have rearranged the sentence to perhaps make it clearer that it is this activation gate that is open, as is the common nomenclature for sodium channel open and closed states. The SF "gate" the referee refers to is suggested to be closed (collapsed actually) in the "potentially inactivated" state crystals of NavAb and NavRh, which also have closed activation gates, but this is not what is referred to commonly when open/closed states are discussed, in part because it may be a structural artefact, unrelated to the functional opening. The "preopen state" of Payandeh et al (2011) has an open SF and a closed activation gate. It is expected that at 0 mV, as is present

in the crystals, the pore will be in an activated state, as seen in these crystals. Some MD studies have suggested that the gate is closed in all the crystal structures to date, but that is clearly not true from the size of the opening in this structure at the intracellular side, which is more than sufficient to pass sodium ions. Again, we must emphasise that this conclusion was based on an experimental structure as opposed to MD calculations.

Furthermore, Naylor et al claim that sodium ions are capable of binding to several closely-related sites as they pass through the selectivity filter ‚possibly due to the presence of intervening water molecules along the length of the selectivity filter'. This claim lacks sufficient evidence.

This statement is based on the evidence provided in this manuscript of the slightly different positions of the ions in the AB and CD dimers, despite the small RMSD between the protein structures in the two crystallographically-distinct tetramers. We have clarified this in the manuscript. In addition to the overlaid different ion sites for the two tetramers shown in Figure 2B, we now show both the ion and water sites in the two tetramers in new Appendix Figure S10.

Apart from a higher structural resolution a molecular dynamics study of the passage of sodium ions through the selectivity filter would be useful.

Several molecular dynamics studies (include Ulmschneider et al) have already been published and cited in this manuscript. Doing an additional MD study as part of this work would seem to be without purpose, and in any case is beyond the scope of this experimentally-based study.

In the second part of their ms Naylor et al study the effect of mutating pore residue E178 to aspartic acid (E178D). The mutation appears to alter the first sodium binding site (NaI) and has dramatic effects on sodium channel properties including single channel conductance, selectivity, current density. This may indicate considerable structural differences between wild type and mutant channel. For example, the mutation may affect the network behind the selectivity filter that stabilizes the pore structure. It is important to provide this information.

The referee is correct, there could be structural differences elsewhere, but in this case the remainder of the pore is unaltered from the wildtype: The all-atom protein RMSD between the WT and mutant structures is only 0.778 Å (and just 0.478 Å over the 40 residues (150-190) in the external leaflet of the pore (tops of S5 and S6 and the intervening SF and pore helices which make up the selectivity filter and its surroundings protein residues). There are no significant changes observable anywhere else in the difference maps, other than in the 178 residue plus the adjacent ion in the SF.

The referee is also correct that we should have included this information in the manuscript, which we have now done in the section on the mutant structure.

In addition, as a demonstration for the referee (although we don't think this is necessary for publication, it could be included as an appendix figure, if the editor deemed appropriate) of how close the wild type and mutant structures are, below is the

Initial $2F_o - F_c$ map for E53D mutant, contoured at +1.0 rmsd in sand. Protein drawn as sticks and sodium ions drawn as spheres, front chain removed for clarity.

It can be seen that the wildtype structure (waters not included) fits into the initial mutant map (before refinement) without any significant differences evident.

This kind of mutation can have dramatic effects for potassium channel pore structures.

Yes it can but it doesn't in this case, which is why it is a good validation for the effects of only that one residue on the ion binding. This has now been noted in the text.

The study is potentially very interesting to a wider audience as it discusses an important aspect of sufficient sodium ion transport across the plasma membrane. However, the results shown provide in evidence for the claims made. Probably for this reason the results section contains many very speculative statements, which should be moved to Discussion.

We have kept the Discussion brief, as per the style of the EMBO J, and when we discuss any speculations (not, as suggested by the referee "many very speculative") we clearly label them as such, and provide additional evidence in the supplementary material.

Minor (or major?) point: Who is Antonella. Please, give the full name. Also, indicate whose present address is at the institute of Molecular Biology and Pathology in Rome.

This was a typographical error that has been corrected in the revised version.

Referee #4:

*Naylor et al report the structure of the ion-selective pore of a bacterial sodium channel with bound sodium ions. Bacterial sodium channels are important model systems for the eukaryotic channels, for which there are no high-resolution structures. A number of structures of bacterial sodium channels have been reported, not least by the Birkbeck group, but none of them have revealed the position of the bound sodium ions in the channel, **which is important for understanding how it works and how ions are selected. The comparison to the ion-bound selectivity filter of potassium channels is particularly instructive, as it shows that potassium ions are desolvated for passage through the channel, whereas sodium ions are not. This appears to be a sufficient, simple explanation of ion selectivity. The observation that the E178D mutant has two instead of three sodium ions is gratifying and reassuring.***

***This is an interesting, succinct and timely manuscript.** Nevertheless there are a few points that merit further attention.*

1. One query relates to the electron density in the channel and its interpretation as sodium ions or water molecules. As the authors point out, these are difficult to tell apart in an x-ray structure at this resolution. Yet whether the channel contains three or four sodium ions is an important point. The densities in the channel are surprisingly low, as they are contoured at 2.5 or even 2.2 sigma.

Actually 2.5 sigma maps are compatible with high or full occupancy (5 sigma would be very unusual for atoms with this number of electrons).

This suggests that the occupancy is low - at full occupancy, sodiums should show up at 5 sigma or higher. What could be the reason for the weak density and low sodium occupancy?

The ions were refined at full occupancy (see new Appendix Figure S10) with the waters at 0.5 occupancy; when doing this the B-factors were very similar to those of the protein atoms. This is further discussed and the effects of different types of changes demonstrated in Appendix Figure S2 and the supplementary discussion section in the appendix. We do acknowledge that occupancy refinement is complicated because the ions lie on asymmetry axis. However, as additional evidence of the occupancy level, sodium ions contain roughly the same number of electrons as the protein backbone atoms and the waters, and so all should appear at around the same density levels, as seen here.

2. The density assigned to a water molecule in the S3 position is even weaker, which must make it hard to be sure it is not a low-occupancy sodium ion, especially since it seems to be coordinated by four water molecules with the same geometry as the sodium ion in S4, as far as I can tell from Figure 2A.

The assignment of the sodium ion in the S3 position was based on geometry (distance between ions and water) as well as the occupancy/density, as discussed in the main text.

Have the occupancies of the sodium ions and water molecules been refined? What are they?

The occupancies were held constant, whilst the B-factors were allowed to change during refinement. For clarity, the occupancies, B-factors and identities of the refined ions and water are listed in the new Appendix Figure 10.

Omit maps for the sodiums and the supposed water molecule on the channel axis should be shown. Unless the authors can rule out that there are actually four sodium ions bound, this should at least be discussed as a possibility, to be on the safe side. **It will not take anything away from the significance of the paper.**

We note that the referee is correct: the peak on the symmetry axis in the AB tetramer assigned as being water can not be absolutely discounted as being a lower occupancy sodium ion. The more conservative interpretation has been to model this density as water, as the electron density at this location is at lower sigma than that for the sodium ion sites and does not appear in the initial difference maps or the omit map. However, it is possible this site is occasionally occupied by a sodium ion, and in that case there could actually be four sodium ions rather than three in the SF.

We now include omit maps as suggested (new Appendix Figure S9), and also have modified the text as suggested, to indicate the possibility suggested by the referee.

3. Are there any conformational changes in the selectivity filter with and without bound sodium? What are they?

As noted above for referee 2, we agree we should have made this point clearer in the manuscript, and now have corrected it accordingly. The only significant changes are in the side chain of residue 178, and in the density attributed to the top sodium ion. The all-atom RMSD (minus the side chain of E178), as noted above, is only 0.778 Å.

4. The authors state that dehydrated potassium ions are of about the right size to pass through the channel, but that this would be energetically unfavourable. How about hydrated potassium ions? Are they too large, and by how much? A table of ionic radii of hydrated and dehydrated sodium, potassium and calcium ions (with error margins!) would be instructive.

The values in the literature vary considerably for ionic and hydrated ionic radii, depending how the value is obtained and the large range of different geometries and coordination numbers cited (see below for a typical survey of these numbers – especially note the wide variation in values used in MD simulations!). The experimentally determined solvation spheres for the free ions in solute are especially large for the smaller ions: this is because their greater charge density results in the presence of second and third hydration spheres. These additional loosely associated spheres of water are not relevant when considering an ion passing through the pore, when only the tightest bound waters will remain associated with the central ion, a fact recognised by the wide range of radii used by MD simulations. Perhaps the clearest indicator, therefore is the directly measured metal-water distances, which show an increase in size with atomic number, and therefore an increase in the size of the first hydration sphere with atomic number for each group of metals.

We are thus reluctant to include the values in this publication (as it is not our own work and we cannot verify the correctness of the values). The metal-water distances for potassium ions are too large to fit in the pore. Hence only partially hydrated sodium and lithium ions or dehydrated ions (too energetically costly for K, Ca and Mg) are potential permeants. Because of the uncertainty of these values, we deem it more prudent to just simply suggest that potassium ions would need to be at least partially dehydrated.

Metal	Covalent radius (Å)	Metal-water distance in proteins (Å)	Hydrated ion radius as used in MD (Å)	Hydrated ionic radii in water ¹
Li	0.6	1.9-2.17 (not in protein)	3.82-1.52	3.4 ²
Na	0.95	2.42	1.04-3.30	2.76
K	1.33	2.84	2.32-2.35	2.32
Ca	0.99	2.39	4.2-1.97	4.28
Mg	0.65	2.07	4.4-1.60	4.12

¹Taken from Nightingale, ER, J. Phys. Chem. **63**, 1381-89 (1959)

²includes second hydration sphere due to electropositivity

5. It would be interesting to know why the crystals now diffract better than before. Is it possible to say what made the difference?

As noted above in the reply to referee 2 (minor corrections), it was many very small changes, none of which made a big change, but all of which contributed to the improvement, over the course of 3.5 years of effort. This is not uncommon for membrane proteins.

Minor points:

6. p5, top para: "... suggesting that protein-ion interactions...". 2nd para: "Thus sodium ions appear..." - check for similar mistakes.

This has been corrected in the revised manuscript.

7. The name of one of the authors is incomplete.

This has been corrected in the revised manuscript.

2nd Editorial Decision

15 January 2016

Thank you for submitting the revised version of your manuscript. It has now been seen by two of the original referees, whose comments are enclosed below.

As you will see they both find that all concerns have been sufficiently addressed and they now recommend the manuscript for publication. However, before we can officially accept the manuscript there are a few editorial issues concerning text and figures that I need you to address,

-> Please provide the main text as a simple text file excluding figures.

-> Please re-check and add in-text references for Appendix Figures S7 and S8.

-> Please upload the main figures as individual figure files (jpg, gif or PDF format, uploaded as "Source data files") in sufficient resolution.

-> Please include a table of contents at the beginning of the appendix in PDF format.

Thank you again for giving us the chance to consider your manuscript for The EMBO Journal, I look forward to your final revision.

REFEREE REPORTS:

Referee #2:

The authors well addressed our Referees' comments. As especially they solved the problems about resolution and statistics, this referee has no experiments to be added. This is the first report to show the sodium recognition mechanism by Nav at high resolution, and this paper should be published in EMBO J.

Referee #4:

No additional comments.